# AlberDICE: Addressing Out-Of-Distribution Joint Actions in Offline Multi-Agent RL via Alternating Stationary Distribution Correction Estimation

**Daiki E. Matsunaga** [*]
KAIST
dematsunaga@ai.kaist.ac.kr

**Jongmin Lee** [*]
UC Berkeley
jongmin.lee@berkeley.edu

**Jaeseok Yoon**
KAIST
jsyoon@ai.kaist.ac.kr

**Stefanos Leonardos**
King's College London
stefanos.leonardos@kcl.ac.uk

**Pieter Abbeel**
UC Berkeley
pabbeel@cs.berkeley.edu

**Kee-Eung Kim**
KAIST
kekim@kaist.ac.kr

## Abstract

One of the main challenges in offline Reinforcement Learning (RL) is the distribution shift that arises from the learned policy deviating from the data collection policy. This is often addressed by avoiding out-of-distribution (OOD) actions during policy improvement as their presence can lead to substantial performance degradation. This challenge is amplified in the offline Multi-Agent RL (MARL) setting since the joint action space grows exponentially with the number of agents. To avoid this curse of dimensionality, existing MARL methods adopt either value decomposition methods or fully decentralized training of individual agents. However, even when combined with standard conservatism principles, these methods can still result in the selection of OOD joint actions in offline MARL. To this end, we introduce AlberDICE, an offline MARL algorithm that alternatively performs centralized training of individual agents based on stationary distribution optimization. AlberDICE circumvents the exponential complexity of MARL by computing the best response of one agent at a time while effectively avoiding OOD joint action selection. Theoretically, we show that the alternating optimization procedure converges to Nash policies. In the experiments, we demonstrate that AlberDICE significantly outperforms baseline algorithms on a standard suite of MARL benchmarks.

## 1 Introduction

Offline Reinforcement Learning (RL) has emerged as a promising paradigm to train RL agents solely from pre-collected datasets [13, 16]. Offline RL aims to address real-world settings in which further interaction with the environment during training is dangerous or prohibitively expensive, e.g., autonomous-car driving, healthcare operations or robotic control tasks [3, 4, 39]. One of the main challenges for successful offline RL is to address the distribution shift that arises from the difference between the policy being learned and the policy used for data collection. Conservatism is a commonly adopted principle to mitigate the distribution shift, which prevents the selection of OOD actions via conservative action-value estimates [11] or direct policy constraints [6].

However, avoiding the selection of OOD actions becomes very challenging in offline Multi-Agent RL (MARL)[2], as the goal is now to stay close to the states and *joint* actions in the dataset. This is

---

[*]Equal Contribution

[2]We assume cooperative MARL for this paper in which agents have common rewards.

37th Conference on Neural Information Processing Systems (NeurIPS 2023).

not trivial since the joint action space scales exponentially with the number of agents, a problem known as the *curse of dimensionality*. Previous attempts to address these issues include decomposing the joint action-value function under strict assumptions such as the Individual-Global-Max (IGM) principle [30, 33, 36, 40], or decentralized training which ignores the non-stationarity caused by the changing policies of other agents [18, 25, 38]. While effective in avoiding the curse of dimensionality, these assumptions are insufficient in avoiding OOD joint action selection even when applying the conservatism principles.

To illustrate the problem of joint action OOD, consider the XOR game in Figure 1. In this game, two agents need to coordinate to achieve optimal joint actions, here, either $(A, B)$ or $(B, A)$. Despite its simple structure, the co-occurrence of two global optima causes many existing algo-

rithms to degenerate in the XOR game [5]. To see this, suppose we have an offline dataset $D = \{(A, A, 0), (A, B, 1), (B, A, 1)\}$. In this situation, IGM-based methods [40] represent the joint $Q(a_1, a_2)$ as a combination of individual $Q_1(a_1)$ and $Q_2(a_2)$, where action $B$ is incentivized over action $A$ by both agents in the individual $Q$ functions. As a consequence, IGM-based methods end up selecting $(B, B)$, which is the OOD joint action. Similarly,

|   | $A$ | $B$ |
|---|-----|-----|
| $A$ | 0 | 1 |
| $B$ | 1 | 0 |

Figure 1: XOR Game

decentralized training methods [25] also choose the OOD joint action $(B, B)$, given that each agent assumes that another agent is fixed with a data policy of selecting $A$ with probability $\frac{2}{3}$. Furthermore, we can see that even behavior-cloning on the expert-only dataset, i.e., $D = \{(A, B), (B, A)\}$, may end up selecting OOD joint actions as well: each individual policy $\pi_1(a_1)$ and $\pi_2(a_2)$ will be uniform over the two individual actions, leading to uniform action selection over the entire joint action space; thus, both $(A, A)$ and $(B, B)$ can be selected. Consequently, OOD joint actions can be hard to avoid especially in these types of environments with multiple global optima and/or when the offline dataset consists of trajectories generated by a mixture of data collection policies.

**Our approach and results**   To address these challenges, we introduce *AlberDICE (ALternate BEst Response Stationary DIstribution Correction Estimation)*, a novel offline MARL algorithm for avoiding OOD actions in the joint action space while circumventing the curse of dimensionality. We start by presenting a coordinate descent-like training procedure where each agent sequentially computes their best response policy while fixing the policies of others. In order to do this in an offline manner, we utilize the linear programming (LP) formulation of RL for optimizing stationary distribution, which has been adapted in offline RL [14] as a stable training procedure where value estimations of OOD actions are eliminated. Furthermore, we introduce a regularization term to the LP objective which matches the stationary distributions of the dataset in the *joint action space*. This regularization term allows AlberDICE to avoid OOD joint actions as well as the curse of dimensionality without any restrictive assumptions such as factorization of value functions via IGM or fully decentralized training. Overall, our training procedure only requires the mild assumption of Centralized Training and Decentralized Execution (CTDE), a popular paradigm in MARL [17, 28, 30] where we assume access to all global information such as state and joint actions during training while agents act independently during execution.

Theoretically, we show that our regularization term preserves the common reward structure of the underlying task and that the sequence of generated policies converges to a Nash policy (Theorem 4.2). We also conduct extensive experiments to evaluate our approach on a standard suite of MARL environments including the XOR Game, Bridge [5], Multi-Robot Warehouse [27], Google Research Football [12] and SMAC [31], and show that AlberDICE significantly outperforms baselines. To the best of our knowledge, AlberDICE is the first DICE-family algorithm successfully applied to the MARL setting while addressing the problem of OOD joint actions in a principled manner.[3]

## 2   Background

**Multi-Agent MDP (MMDP)**   We consider the fully cooperative MARL setting, which can be formalized as a Multi-Agent Markov Decision Process (MMDP) [24] [4]. An $N$-Agent MMDP is defined by a tuple $G = \langle \mathcal{N}, \mathcal{S}, \mathcal{A}, r, P, p_0, \gamma \rangle$ where $\mathcal{N} = \{1, 2, \ldots, N\}$ is the set of agent indices, $s \in \mathcal{S}$ is the state, $\mathcal{A} = \mathcal{A}_1 \times \cdots \times \mathcal{A}_N$ is the joint action space, $p_0 \in \Delta(\mathcal{S})$ is the initial state

---

[3]Our code is available at `https://github.com/dematsunaga/alberdice`

[4]We consider MMDPs rather than Dec-POMDP for simplicity. However, our method can be extended to the Dec-POMDP settings as shown in the experiments. We provide further details in Appendix J.1

distribution, and $\gamma \in (0,1)$ is the discount factor. At each time step, every agent selects an action $a_i \in \mathcal{A}_i$ via a policy $\pi_i(a_i|s)$ given a state $s \in \mathcal{S}$, and receives a reward according to a joint (among all agents) reward function $r : \mathcal{S} \times \mathcal{A} \to \mathbb{R}$. Then, the state is transitioned to the next state $s'$ via the transition probability $P : \mathcal{S} \times \mathcal{A} \to \Delta(\mathcal{S})$ which depends on the global state $s \in S$ and the joint action $\mathbf{a} = \{a_1, \ldots, a_N\} \in \mathcal{A}$ and the process repeats. We also use $\mathbf{a}_{-i}$ to denote the actions taken by all agents other than $i$.

Given a joint policy $\boldsymbol{\pi} = \{\pi_1, \ldots, \pi_N\}$ $\big($or $\boldsymbol{\pi} = \{\pi_i, \boldsymbol{\pi}_{-i}\}$ when we single out agent $i \in \mathcal{N}\big)$, we have that $\boldsymbol{\pi}(\mathbf{a}|s) = \pi_1(a_1|s) \cdot \pi_2(a_2|s, a_1) \cdots \pi_N(a_N|s, a_1, \ldots, a_{N-1})$. If the $\pi_i$'s can be factorized into a product of individual policies, then $\boldsymbol{\pi}$ is *factorizable*, and we write $\boldsymbol{\pi}(\mathbf{a}|s) = \prod_{j=1}^{N} \pi_j(a_j|s)$. We denote the set of all policies as $\Pi$, and the set of factorizable policies as $\Pi_f$. Given a joint policy $\boldsymbol{\pi}$, a state value function $V^{\boldsymbol{\pi}}$ and a action-value function $Q^{\boldsymbol{\pi}}$ are defined by:

$$V^{\boldsymbol{\pi}}(s) := \mathbb{E}_{\boldsymbol{\pi}} \left[ \sum_{t=0}^{\infty} \gamma^t r(s_t, \mathbf{a}_t)|s_0 = s \right], \quad Q^{\boldsymbol{\pi}}(s, \mathbf{a}) := \mathbb{E}_{\boldsymbol{\pi}} \left[ \sum_{t=0}^{\infty} \gamma^t r(s_t, \mathbf{a}_t)|s_0 = s, \mathbf{a}_0 = \mathbf{a} \right]$$

The goal is to find a *factorizable* policy $\pi \in \Pi_f$ that maximizes the joint rewards so that each agent can act in a decentralized manner during execution. We will consider solving MMDP in terms of optimizing stationary distribution. For a given policy $\boldsymbol{\pi}$, its stationary distribution $d^{\boldsymbol{\pi}}$ is defined by:

$$d^{\boldsymbol{\pi}}(s, \mathbf{a}) = (1 - \gamma) \sum_{t=0}^{\infty} \gamma^t \Pr(s_t = s, \mathbf{a}_t = \mathbf{a}), \quad s_0 \sim p_0, \ \mathbf{a}_t \sim \boldsymbol{\pi}(s_t), \ s_{t+1} \sim P(s_t, \mathbf{a}_t), \ \forall t \geq 0.$$

In offline MARL, online interaction with the environment is not allowed, and the policy is optimized only using the offline dataset $D = \{(s, \mathbf{a}, r, s')_k\}_{k=1}^{|D|}$ collected by diverse data-collection agents. We denote the dataset distribution as $d^D$ and abuse the notation $d^D$ for $s \sim d^D$, $(s, a) \sim d^D$, and $(s, a, s') \sim d^D$.

**Nash Policy** We focus on finding the factorizable policy $\{\pi_i : \mathcal{S} \to \Delta(\mathcal{A}_i)\}_{i=1}^{N} \in \Pi_f$ during centralized training, which necessitate the notions of Nash and $\epsilon$-Nash policies.

**Definition 2.1** (Nash and $\epsilon$-Nash Policy). A joint policy $\boldsymbol{\pi}^* = \langle \pi_i^* \rangle_{i=1}^{N}$ is a *Nash policy* if it holds

$$V_i^{\pi_i^*, \boldsymbol{\pi}_{-i}^*}(s) \geq V_i^{\pi_i, \boldsymbol{\pi}_{-i}^*}(s), \quad \forall i \in \mathcal{N}, \ \pi_i, \ s \in \mathcal{S}. \tag{1}$$

Similarly, a joint policy $\boldsymbol{\pi}^* = \langle \pi_i^* \rangle_{i=1}^{N}$ is an $\epsilon$-*Nash policy* if there exists an $\epsilon > 0$ so that for each agent $i \in \mathcal{N}$, it holds that $V_i^{\pi_i^*, \boldsymbol{\pi}_{-i}^*}(s) \geq V_i^{\pi_i, \boldsymbol{\pi}_{-i}^*}(s) - \epsilon$, for all $\pi_i, s$. In other words, $\boldsymbol{\pi}^* = \langle \pi_i^*, \boldsymbol{\pi}_{-i}^* \rangle$ is a Nash policy, if each agent $i \in \mathcal{N}$ has no incentive to deviate from $\pi_i^*$ to an alternative policy, $\pi_i$, given that all other agents are playing $\boldsymbol{\pi}_{-i}^*$.

# 3 AlberDICE

In this section, we introduce AlberDICE, an offline MARL algorithm that optimizes the stationary distribution of each agent's factorizable policy while effectively avoiding OOD joint actions. AlberDICE circumvents the exponential complexity of MARL by computing the best response of one agent at a time in an alternating manner. In contrast to existing methods that adopt *decentralized* training of each agent [25] where other agents' policies are assumed to follow the dataset distribution, AlberDICE adopts *centralized* training of each agent, which takes into account the non-stationarity incurred by the changes in other agents' policies.

**Regularized Linear Program (LP) for MMDP** The derivation of our algorithm starts by augmenting the standard linear program for MMDP with an additional KL-regularization term, where only a single agent $i \in \mathcal{N}$ is being optimized while other agents' policies $\boldsymbol{\pi}_{-i}$ are fixed:

$$\max_{d_i \geq 0} \sum_{s, a_i, \mathbf{a}_{-i}} d_i(s, a_i) \boldsymbol{\pi}_{-i}(\mathbf{a}_{-i}|s) r(s, a_i, \mathbf{a}_{-i}) - \alpha \, \mathrm{D}_{\mathrm{KL}} \left( d_i(s, a_i) \boldsymbol{\pi}_{-i}(\mathbf{a}_{-i}|s) \| d^D(s, a_i, \mathbf{a}_{-i}) \right) \tag{2}$$

$$\text{s.t.} \sum_{a_i', \mathbf{a}_{-i}'} d_i(s', a_i') \boldsymbol{\pi}_{-i}(\mathbf{a}_{-i}'|s') = (1 - \gamma) p_0(s') + \gamma \sum_{s, a_i, \mathbf{a}_{-i}} P(s'|s, a_i, \mathbf{a}_{-i}) d_i(s, a_i) \boldsymbol{\pi}_{-i}(\mathbf{a}_{-i}|s) \ \forall s', \tag{3}$$

where $\mathrm{D}_{\mathrm{KL}}\left(p(x)\|q(x)\right) := \sum_x p(x) \log \frac{p(x)}{q(x)}$ is the KL-divergence between probability distributions $p$ and $q$, and $\alpha > 0$ is a hyperparameter that controls the degree of conservatism, i.e., the amount of

penalty for deviating from the data distribution, which is a commonly adopted principle in offline RL [8, 14, 22, 42]. Satisfying the Bellman-flow constraints (3) guarantees that $d(s, a_i, \mathbf{a}_{-i}) := d_i(s, a_i)\boldsymbol{\pi}_{-i}(\mathbf{a}_{-i}|s)$ is a valid stationary distribution in the MMDP.

As we show in our theoretical treatment of the regularized LP in Section 4, the selected regularization term defined in terms of joint action space critically ensures that every agent $i \in \mathcal{N}$ optimizes the *same* objective function in (2). This ensures that when agents optimize alternately, the objective function always monotonically improves which, in turn, guarantees convergence (see Theorem 4.2). This is in contrast to existing methods such as [25], where each agent optimizes the *different* objective functions. Importantly, this is achieved while ensuring conservatism. As can be seen from (2), the KL-regularization term is defined in terms of the *joint* stationary distribution of *all* agents which ensures that the optimization of the regularized LP effectively avoids OOD joint action selection.

The optimal solution of the regularized LP (2-3), $d_i^*$, corresponds to the stationary distribution for a best response policy $\pi_i^*$ against the fixed $\boldsymbol{\pi}_{-i}$, and $\pi_i^*$ can be obtained by $\pi_i^* = \frac{d_i^*(s,a_i)}{\sum_{a_i'} d_i^*(s,a_i)}$.

The (regularized) LP (2-3) can also be understood as solving a (regularized) *reduced MDP* $\bar{M}_i = \langle \mathcal{S}, \mathcal{A}_i, \bar{P}_i, \bar{r}_i, \gamma, p_0 \rangle$ for a single agent $i \in \mathcal{N}$, where $\bar{P}_i$ and $\bar{r}_i$ are defined as follows[5]:

$$\bar{P}_i(s'|s, a_i) := \sum_{\mathbf{a}_{-i}} \boldsymbol{\pi}_{-i}(\mathbf{a}_{-i}|s)P(s'|s, a_i, \mathbf{a}_{-i}), \quad \bar{r}_i(s, a_i) := \sum_{\mathbf{a}_{-i}} \boldsymbol{\pi}_{-i}(\mathbf{a}_{-i}|s)r(s, a_i, \mathbf{a}_{-i}).$$

Then, $d_i^*$ is an optimal stationary distribution on the reduced MDP, $\bar{M}_i$, but the reduced MDP is non-stationary due to other agents' policy, $\boldsymbol{\pi}_{-i}$, updates. Therefore, it is important to account for changes in $\boldsymbol{\pi}_{-i}$ during training in order to avoid selection of OOD joint actions.

**Lagrangian Formulation**  The constrained optimization (2-3) is not directly solvable since we do not have a white-box model for the MMDP. In order to make (2-3) amenable to offline learning in a model-free manner, we consider a Lagrangian of the constrained optimization problem:

$$\min_{\nu_i} \max_{d_i \geq 0} \mathbb{E}_{\substack{(s,a_i)\sim d_i \\ \mathbf{a}_{-i}\sim\pi_i(s)}} [r(s, a_i, \mathbf{a}_{-i})] - \alpha \sum_{s,a_i,\mathbf{a}_{-i}} d_i(s, a_i)\pi_{-i}(\mathbf{a}_{-i}|s) \log \frac{d_i(s,a_i)\pi_{-i}(\mathbf{a}_{-i}|s)}{d^D(s,a_i)\pi^D_{-i}(\mathbf{a}_{-i}|s,a_i)}$$

$$+ \sum_{s'} \nu_i(s')\Big[(1-\gamma)p_0(s') - \sum_{a_i'} d_i(s', a_i') + \gamma \sum_{s,a_i,\mathbf{a}_{-i}} P(s'|s, a_i, \mathbf{a}_{-i})d_i(s, a_i)\pi_{-i}(\mathbf{a}_{-i}|s)\Big] \quad (4)$$

where $\nu_i(s) \in \mathbb{R}$ is the Lagrange multiplier for the Bellman flow constraints[6]. Still, (4) is not directly solvable due to its requirement of $P(s'|s, a_i, \mathbf{a}_{-i})$ for $(s, a_i) \sim d_i$ that are not accessible in the offline setting. To make progress, we re-arrange the terms in (4) as follows

$$\min_{\nu_i} \max_{d_i \geq 0} (1-\gamma)\mathbb{E}_{s_0\sim p_0}[\nu_i(s_0)] + \mathbb{E}_{(s,a_i)\sim d_i}\Big[ -\alpha \log \frac{d_i(s,a_i)}{d^D(s,a_i)} \quad (5)$$

$$+ \mathbb{E}_{\substack{\mathbf{a}_{-i}\sim\boldsymbol{\pi}_{-i}(s) \\ s'\sim P(s,a_i,\mathbf{a}_{-i})}} \underbrace{\Big[r(s, a_i, \mathbf{a}_{-i}) - \alpha \log \frac{\boldsymbol{\pi}_{-i}(\mathbf{a}_{-i}|s)}{\boldsymbol{\pi}^D_{-i}(\mathbf{a}_{-i}|s,a_i)} + \gamma\nu_i(s') - \nu_i(s)\Big]}_{=:e_{\nu_i}(s,a_i)} \Big]$$

$$= \min_{\nu_i} \max_{d_i \geq 0} (1-\gamma)\mathbb{E}_{s_0\sim p_0}[\nu_i(s_0)] + \mathbb{E}_{(s,a_i)\sim d^D}\Big[ \underbrace{\frac{d_i(s,a_i)}{d^D(s,a_i)}}_{=:w_i(s,a_i)}\big(e_{\nu_i}(s, a_i) - \alpha \log \frac{d_i(s,a_i)}{d^D(s,a_i)}\big)\Big] \quad (6)$$

$$= \min_{\nu_i} \max_{w_i \geq 0} (1-\gamma)\mathbb{E}_{s_0\sim p_0}[\nu_i(s_0)] + \mathbb{E}_{(s,a_i)\sim d^D}\big[w_i(s, a_i)\big(e_{\nu_i}(s, a_i) - \alpha \log w_i(s, a_i)\big)\big] \quad (7)$$

where $e_{\nu_i}(s, a_i)$ is the advantage by $\nu_i$, and $w_i(s, a_i)$ are the stationary distribution correction ratios between $d_i$ and $d^D$. Finally, to enable every term in (7) to be estimated from samples in the offline dataset $D$, we adopt importance sampling, which accounts for the distribution shift in other agents' policies, $\boldsymbol{\pi}_{-i}$:

$$\min_{\nu_i} \max_{w_i \geq 0} (1-\gamma)\mathbb{E}_{p_0}[\nu_i(s_0)] +$$

$$+ \mathbb{E}_{(s,a_i,\mathbf{a}_{-i}s')\sim d^D}\Big[w_i(s, a_i)\frac{\boldsymbol{\pi}_{-i}(\mathbf{a}_{-i}|s)}{\boldsymbol{\pi}^D_{-i}(\mathbf{a}_{-i}|s,a_i)}\big(\hat{e}_{\nu_i}(s, a_i, \mathbf{a}_{-i}, s') - \alpha \log w_i(s, a_i)\big)\Big] \quad (8)$$

---

[5]The reduced MDP is also used by [44], where it is termed *averaged* MDP.

[6]The use of $\min_{\nu_i} \max_{d_i \geq 0}$ rather than $\max_{d_i \geq 0} \min_{\nu_i}$ is justified due to the convexity of the optimization problem in (2) which allows us to invoke strong duality and Slater's condition.

where $\hat{e}_{\nu_i}(s, a_i, \mathbf{a}_{-i}, s') := r(s, a_i, \mathbf{a}_{-i}) - \alpha \log \frac{\boldsymbol{\pi}_{-i}(\mathbf{a}_{-i}|s)}{\boldsymbol{\pi}_{-i}^D(\mathbf{a}_{-i}|s, a_i)} + \gamma \nu_i(s') - \nu_i(s)$. Every term in (8) can be now evaluated using only the samples in the offline dataset. Consequently, AlberDICE aims to solve the unconstrained minimax optimization (8) for each agent $i \in \mathcal{N}$. Once we compute the optimal solution $(\nu_i^*, w_i^*)$ of (8), we obtain the information about the optimal policy $\pi_i^*$ (i.e. the best response policy against the fixed $\boldsymbol{\pi}_{-i}$) in the form of distribution correction ratios $w_i^* = \frac{d^{\pi_i^*}(s, a_i)}{d^D(s, a_i)}$.

**Pretraining autoregressive data policy** To optimize (8), we should be able to evaluate $\boldsymbol{\pi}_{-i}^D(\mathbf{a}_{-i}|s, a_i)$ for each $(s, a_i, \mathbf{a}_{-i}) \in D$. To this end, we pretrain the data policy via behavior cloning, where we adopt an MLP-based autoregressive policy architecture, similar to the one in [43]. The input dimension of $\boldsymbol{\pi}_{-i}^D$ only grows linearly with the number of agents. Then, for each $i \in \mathcal{N}$, we optimize the following:

$$\max_{\boldsymbol{\pi}_{-i}^D} \mathbb{E}_{(s, a_i, \mathbf{a}_{-i}) \sim d^D} \left[ \sum_{j=1, j\neq i}^N \log \boldsymbol{\pi}_{-i}^D(a_j|s, a_i, a_{<j}) \right] \tag{9}$$

While, in principle, the joint action space grows exponentially with the number of agents, learning a joint data distribution in an autoregressive manner is known to work quite well in practice [7, 29].

**Practical Algorithm: Minimax to Min** Still, solving the nested minimax optimization (7) can be numerically unstable in practice. In this section, we derive a practical algorithm that solves a single minimization only using offline samples. For brevity, we denote each sample $(s, a_i, \mathbf{a}_{-i}, s')$ in the dataset as $x$. Also, let $\hat{\mathbb{E}}_{x \in D}[f(x)] := \frac{1}{|D|} \sum_{x \in D} f(x)$ be a Monte-Carlo estimate of $\mathbb{E}_{x \sim p}[f(x)]$, where $D = \{x_k\}_{k=1}^{|D|} \sim p$. First, we have an unbiased estimator of (7):

$$\min_{\nu_i} \max_{w_i \geq 0} (1 - \gamma) \hat{\mathbb{E}}_{s_0 \in D_0}[\nu_i(s_0)] + \hat{\mathbb{E}}_{x \in D} \left[ w_i(s, a_i) \rho_i(x) \left( \hat{e}_{\nu_i}(x) - \alpha \log w_i(s, a_i) \right) \right] \tag{10}$$

where $\rho_i(x)$ is defined as:

$$\rho_i(x) := \frac{\boldsymbol{\pi}_{-i}(\mathbf{a}_{-i}|s)}{\boldsymbol{\pi}_{-i}^D(\mathbf{a}_{-i}|s, a_i)} = \frac{\prod_{j \neq i} \pi_j(a_j|s)}{\boldsymbol{\pi}_{-i}^D(\mathbf{a}_{-i}|s, a_i)}. \tag{11}$$

Optimizing (10) can suffer from large variance due to the large magnitude of $\rho(x)$, which contains products of $N - 1$ policies. To remedy the large variance issue, we adopt Importance Resampling (IR) [32] to (10). Specifically, we sample a mini-batch of size $K$ from $D$ with probability proportional to $\rho(x)$, which constitutes a resampled dataset $D_{\rho_i} = \{(s, a_i, \mathbf{a}_{-i}, s')_k\}_{k=1}^K$. Then, we solve the following optimization, which now does not involve the importance ratio:

$$\min_{\nu_i} \max_{w_i \geq 0} (1 - \gamma) \hat{\mathbb{E}}_{s_0 \in D_0}[\nu_i(s_0)] + \bar{\rho}_i \hat{\mathbb{E}}_{x \in D_{\rho_i}} \left[ w_i(s, a_i) \left( \hat{e}_{\nu_i}(x) - \alpha \log w_i(s, a_i) \right) \right] \tag{12}$$

where $\bar{\rho}_i := \hat{\mathbb{E}}_{x \in D}[\rho_i(x)]$. It can be proven that (12) is still an unbiased estimator of (7) thanks to the bias correction term of $\bar{\rho}$ [32]. The resampling procedure can be understood as follows: for each data sample $x = (s, a_i, \mathbf{a}_{-i}, s')$, if other agents' policy $\boldsymbol{\pi}_{-i}$ selects the action $\mathbf{a}_{-i} \in D$ with low probability, i.e., $\boldsymbol{\pi}_{-i}(\mathbf{a}_{-i}|s) \approx 0$, the sample $x$ will be removed during the resampling procedure, which makes the samples in the resampled dataset $D_{\rho_i}$ consistent with the reduced MDP $\bar{M}_i$'s dynamics. Finally, to avoid the numerical instability associated with solving a min-max optimization problem, we exploit the properties of the inner-maximization problem in (12), specifically, its concavity in $w_i$, and derive its closed-form solution.

**Proposition 3.1.** *The closed-form solution for the inner-maximization in (12) for each $x$ is given by*

$$\hat{w}_{\nu_i}^*(x) = \exp\left( \frac{1}{\alpha} \hat{e}_{\nu_i}(x) - 1 \right) \tag{13}$$

By plugging equation (13) into (12), we obtain the following minimization problem:

$$\min_{\nu_i} \bar{\rho}_i \alpha \hat{\mathbb{E}}_{x \in D_{\rho_i}} \left[ \exp\left( \frac{1}{\alpha} \hat{e}_{\nu_i}(x) - 1 \right) \right] + (1 - \gamma) \mathbb{E}_{s_0 \sim p_0}[\nu_i(s_0)] =: L(\nu_i). \tag{14}$$

As we show in Proposition B.1 in the Appendix, $\tilde{L}(\nu_i)$ is an unconstrained convex optimization problem where the function to learn $\nu_i$ is *state-dependent*. Furthermore, the terms in (14) are estimated only using the $(s, a_i, \mathbf{a}_{-i}, s')$ samples in the dataset, making it free from the extrapolation error by bootstrapping OOD action values. Also, since $\nu_i(s)$ does not involve joint actions, it is not required to adopt IGM-principle in $\nu_i$ network modeling; thus, there is no need to limit the expressiveness power of the function approximator. In practice, we parameterize $\nu_i$ using simple MLPs, which take the state $s$ as an input and output a scalar value.

**Policy Extraction**  The final remaining step is to extract a policy from the estimated distribution correction ratio $w_i^*(s, a_i) = \frac{d^{\pi_i^*}(s, a_i)}{d^D(s, a_i)}$. Unlike actor-critic approaches which perform intertwined optimizations by alternating between policy evaluation and policy improvement, solving (14) directly results in the optimal $\nu_i^*$. However, this does not result in an executable policy. We therefore utilize the I-projection policy extraction method from [14] which we found to be most numerically stable

$$\arg \min_{\pi_i} D_{KL}\left(d^D(s)\pi_i(a_i|s)\boldsymbol{\pi}_{-i}(\mathbf{a}_{-i}|s)||d^D(s)\pi_i^*(a_i|s)\boldsymbol{\pi}_{-i}(\mathbf{a}_{-i}|s)\right) \tag{15}$$

$$= \arg \min_{\pi_i} \hat{\mathbf{E}}_{s\in D, a_i \sim \pi_i}\left[-\log w_i^*(s, a_i) + D_{KL}(\pi_i(a_i|s)||\pi_i^D(a_i|s))\right] \tag{16}$$

In summary, AlberDICE computes the best response policy of agent $i$ by: (1) resampling data points based on the other agents' policy ratios $\rho$ (11) where the data policy $\boldsymbol{\pi}_{-i}^D(\mathbf{a}_{-i}|s, a_i)$ can be pretrained, (2) solving a minimization problem to find $\nu_i^*(s)$ (31) and finally, (3) extracting the policy using the obtained $\nu_i^*$ by I-projection (15). In practice, rather than training $\nu_i$ until convergence at each iteration, we perform a single gradient update for each agent $\nu_i$ and $\pi_i$ alternatively. We outline the details of policy extraction (Appendix E.2) and the full learning procedure in Algorithm 1 (Appendix E).

## 4  Preservation of Common Rewards and Convergence to Nash Policies

In the previous sections, AlberDICE was derived as a practical algorithm in which agents alternately compute the best response DICE while avoiding OOD joint actions. We now prove formally that this procedure converges to Nash policies. While it is known that alternating best response can converge to Nash policies in common reward settings [1], it is not immediately clear whether the same result holds for the regularized LP (2-3), and hence the regularized reward function of the environment, preserves the common reward structure of the original MMDP. As we show in Lemma 4.1, this is indeed the case, i.e., the modified reward in (2-3) is shared across all agents. This directly implies that optimization of the corresponding LP yields the same value for all agents $i \in \mathcal{N}$ for any joint policy, $\pi$, with factorized individual policies, $\{\pi_i\}_{i\in\mathcal{N}}$.

**Lemma 4.1.** *Consider a joint policy $\boldsymbol{\pi} = (\pi_i)_{i\in\mathcal{N}}$, with factorized individual policies, i.e., $\boldsymbol{\pi}(\mathbf{a}|s) = \prod_{i\in\mathcal{N}} \pi_i(a_i|s)$ for all $(s, \mathbf{a}) \in S \times \mathcal{A}$ with $\mathbf{a} = (a_i)_{i\in N}$. Then, the regularized objective in the LP formulation of AlberDICE, cf. equation (2), can be evaluated to*

$$\sum_{s, a_i, \mathbf{a}_{-i}} d_i^\pi(s, a_i)\pi_{-i}(\mathbf{a}_{-i}|s)\tilde{r}(s, a_i, \mathbf{a}_{-i}),$$

*with $\tilde{r}(s, a_i, \mathbf{a}_{-i}) := r(s, a_i, \mathbf{a}_{-i}) - \alpha \cdot \log \frac{d^\pi(s)\boldsymbol{\pi}(\mathbf{a}|s)}{d^D(s, a_i, \mathbf{a}_{-i})}$, for all $(s, \mathbf{a}) \in S \times \mathcal{A}$. In particular, for any joint policy, $\boldsymbol{\pi} = (\pi)_{i\in\mathcal{N}}$, with factorized invdividual policies, the regularized objective in the LP formulation of AlberDICE attains the same value for all agents $i \in \mathcal{N}$.*

We can now use Lemma 4.1 to show that AlberDICE enjoys desirable convergence guarantees in tabular domains in which the policies, $\pi_i(a_i|s)$, can be directly extracted from $d_i(s, a_i)$ through the expression $\pi_i(a_i|s) = \frac{d_i(s, a_i)}{\sum_{a_j} d_i(s, a_j)}$.

**Theorem 4.2.** *Given an MMDP, $G$, and a regularization parameter $\alpha \geq 0$, consider the modified MMDP $\tilde{G}$ with rewards $\tilde{r}$ as defined in Lemma 4.1 and assume that each agent alternately solves the regularized LP defined in equations (2-3). Then, the sequence of policy updates, $(\pi^t)_{t\geq 0}$, converges to a Nash policy, $\pi^* = (\pi_i^*)_{i\in\mathcal{N}}$, of $\tilde{G}$.*

The proofs of Lemma 4.1 and Theorem 4.2 are given in Appendix D. Intuitively, Theorem 4.2 relies on the fact that the objectives in the alternating optimization problems (2-3) involve the same rewards for all agents for any value of the regularization parameter, $\alpha \geq 0$, cf. Lemma 4.1. Accordingly, every update by any agent improves this common value function, $(\tilde{V}^\pi(s))_{s\in S}$, and at some point the sequence of updates is bound to terminate at a (local) maximum of $\tilde{V}$. At this point, no agent can improve by deviating to another policy which implies that the corresponding joint policy is a Nash policy of the underlying (modified) MMDP. For practical purposes, it is also relevant to note that the process may terminate at an $\epsilon$-Nash policy (cf. Definition 2.1), since the improvements in the common value function may become arbitrarily small when solving the LPs numerically.

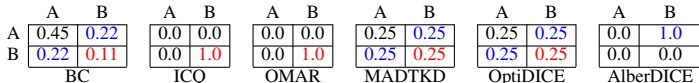

|   | BC |   | ICQ |   | OMAR |   | MADTKD |   | OptiDICE |   | AlberDICE |   |
|---|---|---|---|---|---|---|---|---|---|---|---|---|
|   | A | B | A | B | A | B | A | B | A | B | A | B |
| A | 0.45 | 0.22 | 0.0 | 0.0 | 0.0 | 0.0 | 0.25 | 0.25 | 0.25 | 0.25 | 0.0 | 1.0 |
| B | 0.22 | 0.11 | 0.0 | 1.0 | 0.0 | 1.0 | 0.25 | 0.25 | 0.25 | 0.25 | 0.0 | 0.0 |

Table 1: Policy values after convergence for the Matrix Game in Figure 2 for $D = \{AA, AB, BA\}$

A direct implication from the construction of the AlberDICE algorithm, which is also utilized in the proof of Theorem 4.2, is that the AlberDICE algorithm maintains during its execution and, thus, also returns upon its termination, a factorizable policy, i.e., a policy that can be factorized.

**Corollary 4.3.** *Let $\pi^*$ be the extracted joint policy that is returned from the AlberDICE algorithm. Then, $\pi^*$ is factorizable, i.e., there exist individual policies, $\langle \pi_i^* \rangle_{i \in \mathcal{N}}$, one for each agent, so that $\pi^*(\mathbf{a}|s) = \prod_{i \in \mathcal{N}} \pi_i^*(a_i|s)$ for all $\mathbf{a} \in A, s \in S$.*

## 5  Experimental Results

We evaluate AlberDICE on a series of benchmarks, namely the Penalty XOR Game and Bridge [5], as well as challenging high-dimensional domains such as Multi-Robot Warehouse (RWARE) [27], Google Research Football (GRF) [12] and StarCraft Multi-Agent Challenge (SMAC) [31]. Our baselines include Behavioral Cloning (BC), ICQ [40], OMAR [25], MADTKD [35] and OptiDICE [14] [7]. For a fair comparison, all baseline algorithms use separate network parameters [8] for each agent and the same policy structure. Further details on the dataset are provided in Appendix F.

### 5.1  Penalty XOR Game

We first evaluate AlberDICE on a $2 \times 2$ Matrix Game called Penalty XOR shown in Figure 2. We construct four different datasets: (a) $\{AB\}$, (b) $\{AB, BA\}$, (c) $\{AA, AB, BA\}$, (d) $\{AA, AB, BA, BB\}$. The full results showing the final joint policy values are shown in Table 6 in the Appendix. We show the results for dataset (c) in Table 1. AlberDICE is the only algorithm that converges to a deterministic optimal policy, $BA$ or $AB$ for all datasets.

|   | A | B |
|---|---|---|
| A | 0 | 1 |
| B | 1 | -2 |

Figure 2: Penalty XOR

On the other hand, OptiDICE and MATDKD converges to a stochastic policy where both agents choose $A$ and $B$ with equal probability. This is expected for both algorithms which optimize over joint actions during centralized training which can still lead to joint action OOD if the joint policy is not factorizable. ICQ converges to $AA$ for (b), (d) and $BB$ for (c), which shows the tendency of the IGM constraint and value factorization approaches to converge to OOD joint actions. These results also suggest that the problem of joint action OOD becomes more severe when the dataset collection policy is diverse and/or the environment has multiple global optima requiring higher levels of coordination.

### 5.2  Bridge

Bridge is a stateful extension of the XOR Game, where two agents must take turns crossing a narrow bridge. We introduce a harder version (Figure 3) where both agents start "on the bridge" rather than on the diagonal cells of the opponent goal states as in the original game. This subtle change makes the task much harder because now there are only two optimal actions: (Left, Left) and (Right, Right) at the initial state. Conversely, the original game can be solved optimally as long as at least one agent goes on the bridge.

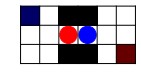

Figure 3: Bridge (Hard)

The *optimal* dataset (500 trajectories) was constructed by a mixture of deterministic optimal policies which randomizes between Agent 1 crossing the bridge first while Agent 2 retreats, and vice-versa. The *mix* dataset further adds 500 trajectories by a uniform random policy.

---

[7]OptiDICE can be naively extended to MARL by training a single $\nu(s)$ network and running Weighted BC as the policy extraction procedure to learn factorized policies, which does not require learning a joint state-action value function. Still, it has some issues (Appendix C).

[8]Fu et al. [5] showed that separating policy parameters are necessary for solving challenging coordination tasks such as Bridge.

Table 2: Mean return and standard error (over 5 random seeds) on the Bridge domain.

| Dataset | | BC | ICQ | OMAR | MADTKD | OptiDICE | AlberDICE |
|---|---|---|---|---|---|---|---|
| Optimal | $-1.26$ | $-2.21 \pm 0.90$ | $-1.81 \pm 0.12$ | $-6.01 \pm 0.00$ | $-4.31 \pm 0.27$ | $-2.71 \pm 0.69$ | $\mathbf{-1.27 \pm 0.03}$ |
| Mix | $-4.56$ | $-5.88 \pm 0.49$ | $-6.01 \pm 0.00$ | $-6.01 \pm 0.00$ | $-6.58 \pm 0.26$ | $-1.76 \pm 0.17$ | $\mathbf{-1.29 \pm 0.00}$ |

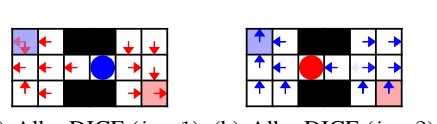 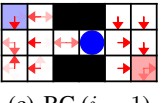 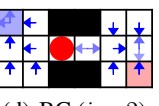

(a) AlberDICE ($i = 1$) (b) AlberDICE ($i = 2$)      (c) BC ($i = 1$)      (d) BC ($i = 2$)

Figure 4: Visualization of learned policies at the initial state for **(from Left to Right) AlberDICE (agent 1, agent 2) and BC (agent 1, agent 2)**, for the Bridge (Hard) on the optimal dataset. The blue arrows indicate agent 2's policies when agent 1 is at ● and red arrows indicate agent 1's policy when agent 2 is at ●.

Table 3: Mean performance and standard error (over 3 random seeds) on the Warehouse domain.

| | Tiny (11x11) | | | Small (11x20) | | |
|---|---|---|---|---|---|---|
| | ($N = 2$) | ($N = 4$) | ($N = 6$) | ($N = 2$) | ($N = 4$) | ($N = 6$) |
| BC | $8.80 \pm 0.25$ | $11.12 \pm 0.19$ | $14.06 \pm 0.32$ | $5.54 \pm 0.06$ | $\mathbf{7.88 \pm 0.14}$ | $8.90 \pm 0.13$ |
| ICQ | $9.38 \pm 0.75$ | $12.13 \pm 0.44$ | $14.59 \pm 0.16$ | $5.43 \pm 0.19$ | $\mathbf{7.93 \pm 0.19}$ | $8.87 \pm 0.22$ |
| OMAR | $6.77 \pm 0.64$ | $\mathbf{14.39 \pm 0.91}$ | $\mathbf{16.13 \pm 1.21}$ | $4.40 \pm 0.34$ | $7.12 \pm 0.38$ | $8.41 \pm 0.49$ |
| MADTKD | $6.24 \pm 0.60$ | $9.90 \pm 0.21$ | $13.06 \pm 0.19$ | $3.65 \pm 0.34$ | $6.85 \pm 0.36$ | $7.85 \pm 0.52$ |
| OptiDICE | $8.70 \pm 0.06$ | $11.13 \pm 0.44$ | $14.02 \pm 0.36$ | $4.84 \pm 0.32$ | $7.68 \pm 0.09$ | $8.47 \pm 0.26$ |
| AlberDICE | $\mathbf{11.15 \pm 0.35}$ | $13.11 \pm 0.32$ | $\mathbf{15.72 \pm 0.36}$ | $\mathbf{5.97 \pm 0.11}$ | $\mathbf{8.18 \pm 0.19}$ | $\mathbf{9.65 \pm 0.13}$ |

Table 4: Mean success rate and standard error (over 5 random seeds) on GRF

| | RPS ($N = 2$) | 3vs1 ($N = 3$) | CA-Hard ($N = 4$) | Corner ($N = 10$) |
|---|---|---|---|---|
| BC | $\mathbf{0.69 \pm 0.08}$ | $\mathbf{0.44 \pm 0.07}$ | $0.70 \pm 0.07$ | $0.24 \pm 0.04$ |
| ICQ | $\mathbf{0.53 \pm 0.39}$ | $0.00 \pm 0.00$ | $0.00 \pm 0.00$ | $0.00 \pm 0.00$ |
| OMAR | $0.00 \pm 0.01$ | $0.01 \pm 0.01$ | $0.00 \pm 0.00$ | $0.07 \pm 0.06$ |
| MADTKD | $\mathbf{0.56 \pm 0.16}$ | $\mathbf{0.56 \pm 0.05}$ | $0.69 \pm 0.05$ | $\mathbf{0.32 \pm 0.07}$ |
| OptiDICE | $\mathbf{0.71 \pm 0.07}$ | $\mathbf{0.50 \pm 0.05}$ | $0.67 \pm 0.15$ | $0.26 \pm 0.04$ |
| AlberDICE | $\mathbf{0.75 \pm 0.12}$ | $\mathbf{0.59 \pm 0.12}$ | $\mathbf{0.83 \pm 0.04}$ | $\mathbf{0.36 \pm 0.04}$ |

The performance results in Table 2 show that AlberDICE can stably perform near-optimally in both the optimal and mix datasets. Also, the learned policy visualizations for the optimal dataset in Figure 4 show that AlberDICE is the only algorithm which converges to the optimal deterministic policy in the initial state, similar to the results in the Matrix game. We also include similar policy visualization results for all states and algorithms in Appendix H.

## 5.3 High-Dimensional MARL Benchmarks

We further evaluate AlberDICE on standard MARL benchmarks including RWARE [26], GRF [12] and SMAC [31]. For RWARE and GRF, we train an autoregressive policy using Multi-Agent Transformers (MAT) [37] in order to collect diverse trajectories for constructing offline datasets. For SMAC, we use the public dataset provided by Meng et al. [19].

RWARE simulates a real-world warehouse in which robots move and deliver requested goods in a partially observable environment (each agent can observe the $3 \times 3$ square centered on the agent). RWARE requires high levels of coordination, especially whenever the density of agents is high and there are narrow pathways where only a single agent can pass through (similar to Bridge).

The results in Table 3 show that AlberDICE performs on-par with OMAR in the Tiny ($11 \times 11$) environment despite OMAR being a decentralized training algorithm. As shown in Figure 9(a) of [26], a large portion of the Tiny map contains wide passageways where agents can move around relatively freely without worrying about colliding with other agents. On the other hand, AlberDICE outperforms baselines in the Small ($11 \times 20$) environment (shown in Figure 9(b) of [26]), where precise coordination among agents becomes more critical since there are more narrow pathways and the probability of a collision is significantly higher. We also note that the performance gap between AlberDICE and baselines is largest when there are more agents ($N = 6$) in the confined space. This increases the probability of a collision, and thus, requires higher levels of coordination.

Table 5: Mean success rate and standard error (over 5 random seeds) on SMAC

| | 3s5z (Hard) (N = 8) | 5m_vs_6m (Hard) (N = 5) | Corridor (SH) (N = 6) | 6hvs8z (SH) (N = 6) | 8m_vs_9m (Hard) (N = 8) | 3s5z_vs_3s6z (SH) (N = 8) |
|---|---|---|---|---|---|---|
| BC | $0.30 \pm 0.05$ | $\mathbf{0.23 \pm 0.02}$ | $0.90 \pm 0.02$ | $0.11 \pm 0.02$ | $0.48 \pm 0.05$ | $0.45 \pm 0.03$ |
| ICQ | $0.18 \pm 0.08$ | $\mathbf{0.18 \pm 0.10}$ | $0.78 \pm 0.03$ | $0.00 \pm 0.00$ | $0.12 \pm 0.21$ | $0.31 \pm 0.04$ |
| OMAR | $\mathbf{0.43 \pm 0.04}$ | $0.18 \pm 0.02$ | $0.92 \pm 0.02$ | $0.15 \pm 0.03$ | $0.45 \pm 0.05$ | $\mathbf{0.60 \pm 0.05}$ |
| MADTKD | $0.12 \pm 0.02$ | $\mathbf{0.19 \pm 0.02}$ | $0.67 \pm 0.01$ | $0.09 \pm 0.02$ | $0.14 \pm 0.04$ | $0.18 \pm 0.02$ |
| OptiDICE | $0.28 \pm 0.05$ | $0.21 \pm 0.02$ | $0.91 \pm 0.02$ | $0.13 \pm 0.00$ | $0.47 \pm 0.05$ | $0.42 \pm 0.04$ |
| AlberDICE | $\mathbf{0.47 \pm 0.03}$ | $\mathbf{0.24 \pm 0.03}$ | $\mathbf{0.98 \pm 0.00}$ | $\mathbf{0.21 \pm 0.03}$ | $\mathbf{0.67 \pm 0.06}$ | $\mathbf{0.63 \pm 0.03}$ |

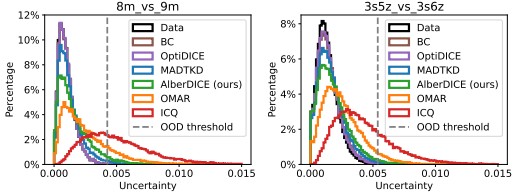

| | 8m_vs_9m (N = 8) | 3s5z_vs_3s6z (N = 8) |
|---|---|---|
| BC | 0.1% | 0.3% |
| ICQ | 54.7% | 26.2% |
| OMAR | 16.5% | 5.7% |
| MADTKD | 1.7% | 0.8% |
| OptiDICE | 0.1% | 0.3% |
| AlberDICE | 4.8% | 1.6% |

(a) Histogram of uncertainty estimates $\{U(s, \mathbf{a}) : s \in D, a \sim \boldsymbol{\pi}(s)\}$.

(b) Percentage of selecting OOD joint actions.

Figure 5: Experimental results to see how effectively AlberDICE avoids OOD joint actions.

Our results for GRF and SMAC in Tables 4 and 5 show that AlberDICE performs consistently well across all scenarios, and outperforms all baselines especially in the Super Hard maps, Corridor and 6h8z. The strong performance by AlberDICE corroborates the importance of avoiding OOD joint actions in order to avoid performance degradation.

### 5.4 Does AlberDICE reduce OOD joint actions?

In order to evaluate how effectively AlberDICE prevents selecting OOD joint actions, we conducted additional experiments on two SMAC domains (8m_vs_9m and 3s5z_vs_3s6z) as follows. First, we trained an uncertainty estimator $U(s, \mathbf{a})$ via fitting random prior [2] $f : S \times \mathcal{A}_1 \times \cdots \times \mathcal{A}_N \to \mathbb{R}^m$ using the dataset $D = \{(s, \mathbf{a})_k\}_{k=1}^{|D|}$. Then, $U(s, \mathbf{a}) = \|f(s, \mathbf{a}) - h(s, \mathbf{a})\|^2$ outputs low values for in-distribution $(s, \mathbf{a})$ samples and outputs high values for out-of-distribution $(s, \mathbf{a})$ samples. Figure 5(a) shows a histogram of uncertainty estimates $U(s, \pi_1(s), \ldots, \pi_N(s))$ for each $s \in D$ and the joint action selected by each method. We set the threshold $\tau$ for determining OOD samples to 99.9%-quantile of $\{U(s, a) : (s, a) \in D\}$. Figure 5(b) presents the percentage of selecting OOD joint actions by each method. AlberDICE selects OOD joint actions significantly **less** often than ICQ (IGM-based method) and OMAR (decentralized training method) while outperforming them in terms of success rate (see Table 5).

## 6 Related Work

**DICE for Offline RL** Numerous recent works utilize the LP formulation of RL to derive DICE algorithms for policy evaluation [20–22]. OptiDICE [14] was introduced as the first policy optimization algorithm for DICE and as a stable offline RL algorithm which does not require value estimation of OOD actions. While OptiDICE can be naively extended to offline MARL in principle, it can still fail to avoid OOD joint actions since its primary focus is to optimize over the joint action space of the MMDP and does not consider the factorizability of policies. We detail the shortcomings of a naive extension of OptiDICE to multi-agent settings in Appendix C.

**Value-Based MARL** A popular method in cooperative MARL is (state-action) value decomposition. This approach can be viewed as a way to model $Q(s, a)$ *implicitly* by aggregating $Q_i$ in a specific manner, e.g., sum [34], or weighted sum [30]. Thus, it avoids modelling $Q(s, a)$ *explicitly* over the joint action space. QTRAN [33] and QPLEX [36] further achieve full representativeness of IGM

[9]. These approaches have been shown to perform well in high-dimensional complex environments including SMAC [31]. However, the IGM assumption and the value decomposition structure have been shown to perform poorly even in simple coordination tasks such as the XOR game [5].

**Policy-Based MARL**   Recently, policy gradient methods such as MAPPO [41] have shown strong performance on many complex benchmarks including SMAC and GRF. Fu et al. [5] showed that independent policy gradient with separate parameters can solve the XOR game and the Bridge environment by converging to a deterministic policy for one of the optimal joint actions. However, it requires an autoregressive policy structure (centralized execution) to learn a stochastic optimal policy which covers multiple optimal joint actions. These empirical findings are consistent with theoretical results [15, 44] showing that running independent policy gradient can converge to a Nash policy in cooperative MARL. On the downside, policy gradient methods are trained with on-policy samples and thus, cannot be extended to the offline RL settings due to the distribution shift problem [16].

**Offline MARL**   ICQ [40] was the first MARL algorithm applied to the offline setting. It proposed an actor-critic approach to overcome the extrapolation error caused by the evaluation of unseen state-action pairs, where the error is shown to grow exponentially with the number of agents. The centralized critic here uses QMIX [30] and thus, it inherits some of the weaknesses associated with value decomposition and IGM. OMAR [25] is a decentralized training algorithm where each agent runs single-agent offline RL over the individual Q-functions and treats other agents as part of the environment. As a consequence, it lacks theoretical motivation and convergence guaranteess in the underlying MMDP or Dec-POMDP. MADTKD [35] extends Multi-Agent Decision Transformers [19] to incorporate credit assignment across agents by distilling the teacher policy learned over the joint action space to each agent (student). This approach can still lead to OOD joint actions since the teacher policy learns a joint policy over the joint action space and the actions are distilled individually to students.

# 7   Limitations

AlberDICE relies on Nash policy convergence which is a well-established solution concept in Game Theory, especially in the general non-cooperative case where each agent may have conflicting reward functions. One limitation of AlberDICE is that the Nash policy may not necessarily correspond to the global optima in cooperative settings. The outcome of the iterative best response depends on the starting point (region of attraction of each Nash policy) and is, thus, generally not guaranteed to find the optimal Nash policy [1]. This is the notorious equilibrium selection problem which is an open problem in games with multiple equilibria, even if they have common reward structure (See Open Questions in [15]). Nonetheless, Nash policies have been used as a solution concept for iterative update of each agents as a way to ensure convergence to factorized policies in Cooperative MARL [10]. Furthermore, good equilibria tend to have larger regions of attraction and practical performance is typically very good as demonstrated by our extensive experiments.

# 8   Conclusion

In this paper, we presented AlberDICE, a multi-agent RL algorithm which addresses the problem of distribution shift in offline MARL by avoiding both OOD joint actions and the exponential nature of the joint action space. AlberDICE leverages an alternating optimization procedure where each agent computes the best response DICE while fixing the policies of other agents. Furthermore, it introduces a regularization term over the stationary distribution of states and joint actions in the dataset. This regularization term preserves the common reward structure of the environment and together with the alternating optimization procedure, allows convergence to Nash policies. As a result, AlberDICE is able to perform robustly across many offline MARL settings, even in complex environments where agents can easily converge to sub-optimal policies and/or select OOD joint actions. As the first DICE algorithm applied to offline MARL with a principled approach to curbing distribution shift, this work provides a starting point for further applications of DICE in MARL and a promising perspective in addressing the main problems of offline MARL.

---

[9]Details about IGM are provided in A.1

## Acknowledgments and Disclosure of Funding

This work was partly supported by the IITP grant funded by MSIT (No.2020-0-00940, Foundations of Safe Reinforcement Learning and Its Applications to Natural Language Processing; No.2022-0-00311, Development of Goal-Oriented Reinforcement Learning Techniques for Contact-Rich Robotic Manipulation of Everyday Objects; No.2019-0-00075, AI Graduate School Program (KAIST); No.2021-0-02068, AI Innovation Hub), NRF of Korea (NRF2019R1A2C1087634), Field-oriented Technology Development Project for Customs Administration through NRF of Korea funded by the MSIT and Korea Customs Service (NRF2021M3I1A1097938), KAIST-NAVER Hypercreative AI Center, the BAIR Industrial Consortium, and NSF AI4OPT AI Centre.

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

# A   Individual-Global-Max (IGM) and its Limitations

Centralized Training with Decentralized Execution (CTDE) refers to a paradigm in MARL where the training phase is permitted to utilize any global information such as the joint policy $\pi = \langle \pi_1, \ldots, \pi_N \rangle$ and global state $s$. Practical CTDE algorithms avoid the combinatorial nature of the joint action space, and reduce the non-stationarity, which arises from agents' simultaneous policy updates during training.

However, an important challenge for CTDE algorithms is that during training they still need to learn a joint policy that can be factorized into individual policies. During the execution phase, agents take individual actions, $a_i$, without conditioning on other agents' actions, $\mathbf{a}_{-i}$, or policies, $\pi_{-i}$. This independence assumption poses a challenge for many pre-existing algorithms, especially for offline MARL when the dataset is generated by a mixture of different data collection policies and/or the environment contains multiple global optima.

One popular way to learn factorizable policies from centralized training is to impose an Individual-Global-Max (IGM) assumption.

**Definition A.1** (Individual-Global-Max (IGM) [30, 33])**.** Individual utility functions $\{Q_i\}_{i=1}^N$ satisfies the IGM condition for a joint state-action value function $Q : S \times \mathcal{A} \to \mathbb{R}$ if the following condition holds:

$$\arg \max_a Q(s, a) = \{\arg \max Q_i(s, a_i)\}_{i=1}^N \tag{17}$$

Intuitively, the IGM condition implies that the optimal Q-function $Q^*$ for a given task or environment can be decomposed into individual utility functions which only condition on the individual actions $a_i$. This assumption results in a space of MARL tasks where decentralized policies (i.e. greedy policies over $Q_i$) can be learned to collectively act optimally. The decomposed utility functions $Q_i$ can then be used for greedy action selection by individual agents [36].

It is easy to see that under the IGM assumption in Definition A.1, we cannot learn a set of individual utility functions $\{Q_i\}$ which can accurately represent the optimal Q-functions of many tasks with multiple global optima, including the XOR game. In fact, [5] showed formally that any algorithm under the IGM constraint cannot represent the underlying optimal Q-function in the XOR game.

# B Proofs for AlberDICE

## B.1 Proof of Proposition 3.1

**Proposition 3.1.** *The closed-form solution for the inner-maximization in (12) for each $x$ is given by*

$$\hat{w}_{\nu_i}^*(x) = \exp\left(\tfrac{1}{\alpha}\hat{e}_{\nu_i}(x) - 1\right) \tag{13}$$

*Proof.* Let

$$L(\nu_i, w_i) := \hat{\mathbf{E}}_{x \in D_{\rho_i}}\left[w_i(s, a_i)\left(\hat{e}_{\nu_i}(s, a_i, \mathbf{a}_{-i}, s') - \alpha \log w_i(s, a_i)\right)\right] + (1-\gamma)\hat{\mathbf{E}}_{s_0 \in D_0}[\nu_i(s_0)]$$

Note that $L(\nu_i, w_i)$ is differentiable and strictly convex for $w_i$. Therefore, we only need to find a point where the gradient becomes zero. For any $x \in D_{\rho_i}$,

$$\frac{\partial L(\nu_i, w_i)}{\partial w_i(x)} = \hat{e}_{\nu_i}(s, a_i, \mathbf{a}_{-i}, s') - \alpha \log w_i(x) - \alpha = 0 \tag{18}$$

$$\iff \hat{w}_i^*(x) = \exp\left(\tfrac{1}{\alpha}\hat{e}_{\nu_i}(s, a_i, \mathbf{a}_{-i}, s') - 1\right) \tag{19}$$

$\square$

## B.2 Convexity of $L(\nu_i)$

**Proposition B.1.** *Let $L(\nu_i)$ be a function, defined as:*

$$L(\nu_i) := \bar{\rho}_i \alpha \hat{\mathbf{E}}_{x \in D_{\rho_i}}\left[\exp\left(\tfrac{1}{\alpha}\hat{e}_{\nu_i}(s, a_i, \mathbf{a}_{-i}, s') - 1\right)\right] + (1-\gamma)\mathbb{E}_{s_0 \sim p_0}[\nu_i(s_0)] \tag{20}$$

*Then, $L(\nu_i)$ is convex with respect to $\nu_i$*

*Proof.* For any functions $\nu_i, \nu_i'$ and constant $\lambda \in [0, 1]$, we can derive the following equality:

$$\hat{e}_{\left(\lambda\nu_i + (1-\lambda)\nu_i'\right)}(s, a_i, \mathbf{a}_{-i}, s')$$

$$= r(s, a_i, \mathbf{a}_{-i}) - \alpha \log \frac{\boldsymbol{\pi}_{-i}(\mathbf{a}_{-i}|s)}{\boldsymbol{\pi}_{-i}^D(\mathbf{a}_{-i}|s, a_i)} + \gamma\left(\lambda\nu_i + (1-\lambda)\nu_i'\right)(s') - \left(\lambda\nu_i + (1-\lambda)\nu_i'\right)(s)$$

$$= r(s, a_i, \mathbf{a}_{-i}) - \alpha \log \frac{\boldsymbol{\pi}_{-i}(\mathbf{a}_{-i}|s)}{\boldsymbol{\pi}_{-i}^D(\mathbf{a}_{-i}|s, a_i)} + \gamma\lambda\nu_i(s') - \lambda\nu_i(s) + \gamma(1-\lambda)\nu_i'(s') - (1-\lambda)\nu_i'(s)$$

$$= \lambda r(s, a_i, \mathbf{a}_{-i}) - \lambda\alpha \log \frac{\boldsymbol{\pi}_{-i}(\mathbf{a}_{-i}|s)}{\boldsymbol{\pi}_{-i}^D(\mathbf{a}_{-i}|s, a_i)} + \gamma\lambda\nu_i(s') - \lambda\nu_i(s)$$

$$+ (1-\lambda)r(s, a_i, \mathbf{a}_{-i}) - (1-\lambda)\alpha \log \frac{\boldsymbol{\pi}_{-i}(\mathbf{a}_{-i}|s)}{\boldsymbol{\pi}_{-i}^D(\mathbf{a}_{-i}|s, a_i)} + \gamma(1-\lambda)\nu_i'(s') - (1-\lambda)\nu_i'(s)$$

$$= \lambda\hat{e}_{\nu_i}(s, a_i, \mathbf{a}_{-i}, s') + (1-\lambda)\hat{e}_{\nu_i'}(s, a_i, \mathbf{a}_{-i}, s').$$

Thus, $e_{\nu_i}$ is the linear function with respect to $\nu_i$. Furthermore, using the convexity of $\exp(\cdot)$,

$$\mathbb{E}_{(s, a_i, \mathbf{a}_{-i}, s') \sim d^D}\left[\frac{\boldsymbol{\pi}_{-i}(\mathbf{a}_{-i}|s)}{\boldsymbol{\pi}_{-i}^D(\mathbf{a}_{-i}|s)} \exp\left(\hat{e}_{\left(\lambda\nu_i + (1-\lambda)\nu_i'\right)}(s, a_i, \mathbf{a}_{-i}, s')\right)\right]$$

$$= \mathbb{E}_{(s, a_i, \mathbf{a}_{-i}, s') \sim d^D}\left[\frac{\boldsymbol{\pi}_{-i}(\mathbf{a}_{-i}|s)}{\boldsymbol{\pi}_{-i}^D(\mathbf{a}_{-i}|s)} \exp\left(\lambda\hat{e}_{\nu_i}(s, a_i, \mathbf{a}_{-i}, s') + (1-\lambda)\hat{e}_{\nu_i'}(s, a_i, \mathbf{a}_{-i}, s')\right)\right]$$

$$\leq \lambda\mathbb{E}_{(s, a_i, \mathbf{a}_{-i}, s') \sim d^D}\left[\frac{\boldsymbol{\pi}_{-i}(\mathbf{a}_{-i}|s)}{\boldsymbol{\pi}_{-i}^D(\mathbf{a}_{-i}|s)} \exp\left(\hat{e}_{\nu_i}(s, a_i, \mathbf{a}_{-i}, s')\right)\right]$$

$$+ (1-\lambda)\mathbb{E}_{(s, a_i, \mathbf{a}_{-i}, s') \sim d^D}\left[\frac{\boldsymbol{\pi}_{-i}(\mathbf{a}_{-i}|s)}{\boldsymbol{\pi}_{-i}^D(\mathbf{a}_{-i}|s)} \exp\left(\hat{e}_{\nu_i'}(s, a_i, \mathbf{a}_{-i}, s')\right)\right]$$

Therefore, $L(\nu_i)$ is convex function with respect to $\nu_i$. $\square$

We also show that our practical algorithm minimizes the upper bound of the original optimization problem in equation (7) restated here:

$$L(\nu_i, w_i) := (1-\gamma)\mathbb{E}_{s_0 \sim p_0}[\nu_i(s_0)] + \mathbb{E}_{(s, a_i) \sim d^D}\left[w_i(s, a_i)\left(e_{\nu_i}(s, a_i) - \alpha \log w_i(s, a_i)\right)\right] \tag{21}$$

**Corollary B.2.** $L(\nu_i)$ *is an upper bound of* $L(\nu_i, w_i^*)$*, where* $w_i^* = \arg\max_{w_i} L(\nu_i, w_i)$*, i.e.,* $L(\nu_i, w_i^*) \le L(\nu_i)$ *always holds, where equality holds when the MDP transition and other agents' policy* $\boldsymbol{\pi}_{-i}$ *are deterministic.*

*Proof.* We first note that the closed-form solution for $\arg\max_{w_i} L(\nu_i, w_i)$ is as follows:

$$w_i^*(s, a_i) = \exp\left(\tfrac{1}{\alpha} e_{\nu_i}(s, a_i) - 1\right). \tag{22}$$

By plugging this into equation (7), we obtain

$$\min_{\nu_i} \alpha \mathbb{E}_{(s,a_i)\sim d^D}\left[\exp\left(\tfrac{1}{\alpha} e_{\nu_i}(s, a_i) - 1\right)\right] + (1-\gamma)\mathbb{E}_{s_0\sim p_0}[\nu_i(s_0)] =: L\left(w_i^*, \nu_i\right)$$

From Jensen's inequality, we get

$$
\begin{aligned}
L\left(w_i^*, \nu_i\right) &= \alpha \mathbb{E}_{(s,a_i)\sim d^D}\left[\exp\left(\tfrac{1}{\alpha} e_{\nu_i}(s, a_i) - 1\right)\right] + (1-\gamma)\mathbb{E}_{s_0\sim p_0}[\nu_i(s_0)] \\
&= \alpha \mathbb{E}_{(s,a_i)\sim d^D}\left[\exp\left(\tfrac{1}{\alpha} \mathbb{E}_{\substack{\mathbf{a}_{-i}\sim\pi_{-i} \\ s'\sim P(s,a_i,\mathbf{a}_{-i})}}[\hat{e}_{\nu_i}(s, a_i, \mathbf{a}_{-i}, s')] - 1\right)\right] + (1-\gamma)\mathbb{E}_{s_0\sim p_0}[\nu_i(s_0)] \\
&\le \alpha \mathbb{E}_{(s,a_i,\mathbf{a}_{-i},s')\sim d^D}\left[\tfrac{\pi_{-i}(\mathbf{a}_{-i}|s)}{\pi_{-i}^D(\mathbf{a}_{-i}|s)}\exp\left(\tfrac{1}{\alpha}\hat{e}_{\nu_i}(s, a_i, \mathbf{a}_{-i}, s') - 1\right)\right] + (1-\gamma)\mathbb{E}_{s_0\sim p_0}[\nu_i(s_0)] \\
&= L(\nu_i) \tag{23}
\end{aligned}
$$

Also, the inequality becomes tight when the transition model and the opponent policies are deterministic, since $\exp\left(\tfrac{1}{\alpha}\mathbb{E}_{\substack{a_{-i}\sim\pi_{-i} \\ s'\sim P(s,a_i,\mathbf{a}_{-i})}}[\hat{e}(s, a_i, \mathbf{a}_{-i}, s')]\right) = \mathbb{E}_{\substack{\mathbf{a}_{-i}\sim\pi_{-i} \\ s'\sim P(s,a_i,\mathbf{a}_{-i})}}\left[\exp\left(\tfrac{1}{\alpha}\hat{e}(s, a_i, a_{-i}, s')\right)\right]$
should always hold for the deterministic transition $P$ and opponent policies $\pi_{-i}$. $\quad\square$

## C   Problems with Naive Extension of OptiDICE to Offline MARL

OptiDICE [14] is a (single-agent) offline policy optimization algorithm which is derived from the regularized Linear Programming (LP) formulation for RL. For a given MDP $\langle S, A, P, R, \gamma \rangle$, the derivation of OptiDICE starts with the regularized dual of the LP:

$$\max_{d \geq 0} \sum_{s,a} d(s,a)r(s,a) - \alpha \, \mathrm{D_{KL}}(d||d^D) \tag{24}$$

$$\text{s.t.} \sum_{a'} d(s',a') = (1-\gamma)p_0(s') + \gamma \sum_{s,a} P(s'|s,a)d(s,a), \forall s'. \tag{25}$$

Here, $d(s,a)$ should be a stationary distribution of some policy $\pi$ by the Bellman flow constraints (25) ($d^\pi(s,a) := (1-\gamma)\sum_{t=0}^{\infty} \gamma^t \Pr(s_t = s, a_t = a; \pi)$), and $d^D$ is the dataset distribution. The goal is to maximize the rewards while not deviating too much from the data distribution, following the conservatism principle in offline RL. Without the regularization term $\mathrm{D_{KL}}(d||d^D)$, the optimal solution of (24-25) is the stationary distribution for the optimal policy, $d^* = d^{\pi^*}$.

One of the main contributions of OptiDICE is a tractable re-formulation of the dual LP problem above into a single convex optimization problem,

$$\min_{\nu}(1-\gamma)\mathbb{E}_{s \sim \mu_0}[\nu(s)] + \mathbb{E}_{(s,a) \sim d^D} \left[ w^*_\nu(s,a)e_\nu(s,a) - \alpha w^*_\nu(s,a) \log w^*_\nu(s,a) \right]. \tag{26}$$

$$\text{where } w^*_\nu(s,a) := \exp\left(\frac{1}{\alpha}\left(r(s,a) + \gamma \mathbb{E}_{s'}[\nu(s')] - \nu(s)\right) - 1\right). \tag{27}$$

Here, $\nu(s) \in \mathbb{R}$ is the Lagrangian multiplier for the Bellman flow constraint (25), and it approaches the optimal state value function $V^*(s)$ as $\alpha \to 0$. Once we obtain the optimal solution of (26), $\nu^*$, it was shown that the stationary distribution corrections of the optimal policy is given by $w^*_{\nu^*}$:

$$w^*_{\nu^*}(s,a) = \exp\left(\frac{1}{\alpha}\left(r(s,a) + \gamma \mathbb{E}_{s'}[\nu^*(s')] - \nu^*(s)\right) - 1\right) = \frac{d^*(s,a)}{d^D(s,a)}. \tag{28}$$

However, its extension to MARL can cause a number of subtle issues. While solving (26) does not suffer from the curse of dimensionality posed in MARL since $\nu(s)$ is a state-dependent function, $\nu^*$ itself is not an executable policy. We therefore should extract a policy from it. However, once we try to learn a parametric function for $w(s,a)$, we encounter a combinatorial space of joint actions. We thus should avoid learning any state-action dependent functions for policy extraction. One feasible way to do so is to perform policy extraction via Weighted Behavior-cloning (WBC):

$$\forall i, \ \max_{\pi_i} \mathbb{E}_{(s,a_i,\mathbf{a}_{-i},s') \sim d^D}\left[\hat{w}^*_{\nu^*}(s,a_i,\mathbf{a}_{-i},s')\log\pi_i(a_i|s)\right] \approx \mathbb{E}_{(s,a_i,\mathbf{a}_{-i},s') \sim d^*}[\log\pi_i(a_i|s)] \tag{29}$$

$$\text{s.t. } \hat{w}^*_{\nu^*}(s,a_i,\mathbf{a}_{-i},s') = \exp\left(\frac{1}{\alpha}\left(r(s,a_i,\mathbf{a}_{-i}) + \gamma\nu^*(s') - \nu^*(s)\right) - 1\right), \tag{30}$$

which corresponds to behavior-cloning of the factorized policy on the state-action visits by the optimal (joint) policy. However, if the optimal joint policy (by $\nu^*$) is a multi-modal distribution, this WBC policy extraction step can result in an arbitrarily bad policy, selecting OOD joint actions. For example, consider the XOR matrix game in Figure 1 with a dataset $D = \{(A,A),(A,B),(B,A)\}$, where the optimal joint policy is given by $\boldsymbol{\pi}^*(a_1 = A, a_2 = B) = \boldsymbol{\pi}^*(a_1 = A, a_2 = B) = \frac{1}{2}$. In this situation, WBC of OptiDICE (29) obtains the factorized policies of $\pi_1(a_1 = A) = \pi_1(a_1 = B) = \frac{1}{2}$ and $\pi_2(a_2 = A) = \pi_2(a_2 = B) = \frac{1}{2}$, which can select *suboptimal* (and OOD) joint actions:

$$\textcolor{red}{\boldsymbol{\pi}(a_1 = A, a_2 = A)} = \pi_1(a_1 = A)\pi_2(a_2 = A) = \frac{1}{4}$$

$$\boldsymbol{\pi}(a_1 = A, a_2 = B) = \pi_1(a_1 = A)\pi_2(a_2 = B) = \frac{1}{4}$$

$$\boldsymbol{\pi}(a_1 = B, a_2 = A) = \pi_1(a_1 = B)\pi_2(a_2 = A) = \frac{1}{4}$$

$$\textcolor{red}{\boldsymbol{\pi}(a_1 = B, a_2 = B)} = \pi_1(a_1 = B)\pi_2(a_2 = B) = \frac{1}{4}$$

This analysis is consistent with our experimental results in Section 5, which demonstrated the failure of OptiDICE in solving the Penalty XOR Game as well as Bridge by converging to sub-optimal OOD joint actions.

# D  Proofs for Section 4

To prove Theorem 4.2, we first need to show that the regularized objective in the LP formulation of AlberDICE, cf. equation (2), preserves the common reward structure of $G$.

**Lemma 4.1.** *Consider a joint policy $\boldsymbol{\pi} = (\pi_i)_{i \in \mathcal{N}}$, with factorized individual policies, i.e., $\boldsymbol{\pi}(\mathbf{a}|s) = \prod_{i \in \mathcal{N}} \pi_i(a_i|s)$ for all $(s, \mathbf{a}) \in S \times \mathcal{A}$ with $\mathbf{a} = (a_i)_{i \in N}$. Then, the regularized objective in the LP formulation of AlberDICE, cf. equation (2), can be evaluated to*

$$\sum_{s, a_i, \mathbf{a}_{-i}} d_i^\pi(s, a_i) \pi_{-i}(\mathbf{a}_{-i}|s) \tilde{r}(s, a_i, \mathbf{a}_{-i}),$$

*with $\tilde{r}(s, a_i, \mathbf{a}_{-i}) := r(s, a_i, \mathbf{a}_{-i}) - \alpha \cdot \log \frac{d^\pi(s)\boldsymbol{\pi}(\mathbf{a}|s)}{d^D(s, a_i, \mathbf{a}_{-i})}$, for all $(s, \mathbf{a}) \in S \times \mathcal{A}$. In particular, for any joint policy, $\boldsymbol{\pi} = (\pi)_{i \in \mathcal{N}}$, with factorized invdividual policies, the regularized objective in the LP formulation of AlberDICE attains the same value for all agents $i \in \mathcal{N}$.*

*Proof.* Recall that the KL-divergence, $\mathrm{D}_{\mathrm{KL}}(p(x)\|q(x))$, between probability distributions $p$ and $q$ is defined as $\mathrm{D}_{\mathrm{KL}}(p(x)\|q(x)) := \sum_x p(x) \log \frac{p(x)}{q(x)}$. Thus, the $\mathrm{D}_{\mathrm{KL}}$ term in the objective of equation (2) can be written as

$$\mathrm{D}_{\mathrm{KL}}\left(d_i^\pi(s, a_i)\pi_{-i}(\mathbf{a}_{-i}|s)\|d^D(s, a_i, \mathbf{a}_{-i})\right) = \sum_{s, a_i, \mathbf{a}_{-i}} d_i^\pi(s, a_i)\pi_{-i}(\mathbf{a}_{-i}|s) \cdot \log \frac{d_i^\pi(s, a_i)\pi_{-i}(\mathbf{a}_{-i}|s)}{d^D(s, a_i, \mathbf{a}_{-i})}.$$

Since the $\pi_i's$ are factorized by assumption, the decomposition $d_i^\pi(s, a_i) = d^\pi(s)\pi_i(a_i|s)$, implies that the numerator in the $\log$-term of the previous expression can be written as

$$d_i^\pi(s, a_i)\pi_{-i}(\mathbf{a}_{-i}|s) = d^\pi(s)\pi_i(a_i|s)\pi_{-i}(\mathbf{a}_{-i}|s) = d^\pi(s)\boldsymbol{\pi}(\mathbf{a}|s).$$

Substituting back in the initial expression of the objective function, we obtain that

$$\sum_{s, a_i, \mathbf{a}_{-i}} d_i^\pi(s, a_i)\pi_{-i}(\mathbf{a}_{-i}|s)r(s, a_i, \mathbf{a}_{-i}) - \alpha \, \mathrm{D}_{\mathrm{KL}}\left(d_i^\pi(s, a_i)\pi_{-i}(\mathbf{a}_{-i}|s)\|d^D(s, a_i, \mathbf{a}_{-i})\right)$$

$$= \sum_{s, a_i, \mathbf{a}_{-i}} d_i^\pi(s, a_i)\pi_{-i}(\mathbf{a}_{-i}|s)r(s, a_i, \mathbf{a}_{-i}) - \alpha \sum_{s, a_i, \mathbf{a}_{-i}} d_i^\pi(s, a_i)\pi_{-i}(\mathbf{a}_{-i}|s) \cdot \log \frac{d^\pi(s)\boldsymbol{\pi}(\mathbf{a}|s)}{d^D(s, a_i, \mathbf{a}_{-i})}$$

$$= \sum_{s, a_i, \mathbf{a}_{-i}} d_i^\pi(s, a_i)\pi_{-i}(\mathbf{a}_{-i}|s) \left[ r(s, a_i, \mathbf{a}_{-i}) - \alpha \log \frac{d^\pi(s)\boldsymbol{\pi}(\mathbf{a}|s)}{d^D(s, a_i, \mathbf{a}_{-i})} \right].$$

Thus, by setting $\tilde{r}(s, a_i, \mathbf{a}_{-i}) := r(s, a_i, \mathbf{a}_{-i}) - \alpha \cdot \log \frac{d^\pi(s)\boldsymbol{\pi}(\mathbf{a}|s)}{d^D(s, a_i, \mathbf{a}_{-i})}$, for all $(s, \mathbf{a}) \in S \times \mathcal{A}$, we obtain the claim. The equality of the last expression for all $i \in \mathcal{N}$ follows now immediately from application of the decomposition $d_i^\pi(s, a_i)\pi_{-i}(\mathbf{a}_{-i}|d) = d^\pi(s)\boldsymbol{\pi}(\mathbf{a}|s)$, on the outer expectation for all agent $i \in \mathcal{N}$. $\square$

*Remark* D.1. In the LP formulation of AlberDICE, cf. equation (2) and (3), we used the notation $d_i(s, a_i)$ rather than $d_i^\pi(s, a_i)$, since, in this case, the variables are $d_i(s, a_i)$ are decision variables that are not a-priori related to any particular policy $\pi_i$. However, once the $d_i(s, a_i)$'s are fixed and translated to a policy, e.g., through the relation $\pi_i(a_i|s) = \frac{d_i(s, a_i)}{\sum_{a_j} d_i(s, a_j)}$ that holds in tabular domains, then, we can apply Lemma 4.1. For the purposes of the alternating optimization procedure of the AlberDICE algorithm, Lemma 4.1 ensures that after each update by any agent $i \in N$, the value of the objective function is the same for each agent. To evaluate the modified utilities, $\tilde{r}(s, a_i, \mathbf{a}_{-i})$, during training, agents need to know both the action, $a_i$, *and* the policy, $\pi_i(a, s)$, from whichi this action drawn for each agent $i \in N$. Thus, this regularization terms exploits to the fullest the knowledge available to agents in the centralized training setting.

To proceed, we can define a modified game, $\tilde{G} = \langle N, S, \mathcal{A}, \tilde{r}, P, \gamma \rangle$ which is the same as the original game in every respect, i.e., it has the same state space, agents, actions, transitions and discount factor, except for the rewards, $r$, which are replaced by the modified rewards $\tilde{r}$ for any given value of the regularization parameter, $\alpha > 0$ (for $\alpha = 0$, we simply have the rewards, $r$, of the original game, $G$). Despite this modification, Lemma 4.1 implies that $\tilde{G}$ still has a common reward structure. This can be used to prove that the AlberDICE algorithm has monotonic updates which eventually converge to a Nash equilibrium of the modified game, $\tilde{G}$. For this part, we focus on tabular domains, in which the policies, $\pi_i(a_i \mid s)$, can be directly extracted from $d_i(s, a_i)$ as $\pi_i(a_i \mid s) = \frac{d_i(s, a_i)}{\sum_{a_j} d_i(s, a_j)}$.

**Theorem 4.2.** *Given an MMDP, $G$, and a regularization parameter $\alpha \geq 0$, consider the modified MMDP $\tilde{G}$ with rewards $\tilde{r}$ as defined in Lemma 4.1 and assume that each agent alternately solves the regularized LP defined in equations (2-3). Then, the sequence of policy updates, $(\pi^t)_{t \geq 0}$, converges to a Nash policy, $\pi^* = (\pi_i^*)_{i \in \mathcal{N}}$, of $\tilde{G}$.*

*Proof of Theorem 4.2.* Let $\pi^0 = \langle \pi_i^0 \rangle_{i \in N}$ denote the initial joint policy and let $\pi^t = \langle \pi_i^t \rangle_{i \in N}$ denote the joint policy after iteration $t \in \mathbb{N}$ in an execution of the AlberDICE algorithm. For $i = 1, \dots, N$, we will also write $\pi_{1:i}^t$ to denote the joint policy at time $t$ after players 1 to $i$ have updated their policies, i.e., $\pi_{1:i}^t = (\pi_1^t, \dots, \pi_i^t, \pi_{i+1}^{t-1}, \dots, \pi_N^{t-1})$. Let $d_i(s, a_i) := d_i^{\pi_{1:i-1}^{t-1}}(s, a_i)$ denote the stationary distribution for agent $i$ before the current optimization by player $i$, and let $d_i^*(s, a_i) := d_i^{\pi_{1:i}^{t-1}}(s, a_i), \pi_i^*(a_i \mid s)$ denote the stationary policy derived as the optimal solution of the LP and the corresponding extracted policy for agent $i$, respectively, after the optimization by agent $i$ at time $t$. Then,

$$\tilde{V}_i^{\pi_{1:i}}(s) = \sum_{s, a_i, \mathbf{a}_{-i}} d_i^*(s, a_i) \pi_{-i}(\mathbf{a}_{-i} \mid s) \tilde{r}(s, a_i, \mathbf{a}_{-i})$$

$$\geq \sum_{s, a_i, \mathbf{a}_{-i}} d_i(s, a_i) \pi_{-i}(\mathbf{a}_{-i} \mid s) \tilde{r}(s, a_i, \mathbf{a}_{-i}) = \tilde{V}_i^{\pi_{1:i-1}}(s),$$

for each state $s \in S$, where we used Lemma 4.1 for the equality of the modified rewards among all agents in $N$. The inequality is strict unless agent $i$ is already using an optimal policy, $\pi_i$, against $\pi_{-i}$. Letting $\tilde{V}$ to denote the common value function, i.e., $\tilde{V}_i \equiv \tilde{V}$ for all agents $i \in N$, then, the previous inequality implies that after the update of agent $i$, all agents have a higher value with the current policy $\pi_{1:i}^t$. Thus, the sequence, $\pi^t$, of joint policies generated by the AlberDICE algorithm results in monotonic updates (increases) in the joint modified value function $\tilde{V}(s), s \in S$. Since, $V$ is bounded (rewards are bounded and discounted by assumption), this implies that also $\tilde{V}$ is bounded and hence, at some point, the updates of the algorithm will reach a local maximum of $V$. Let $\pi^* = (\pi_i^*, \pi_{-i}^*)$ denote the extracted policy at that point. Then, for all agents $i \in N$ and any $\pi_i$, it holds that $\tilde{V}_i^{\pi^*}(s) = \tilde{V}^{\pi^*} \geq \tilde{V}^{(\pi_i, \pi_{-i}^*)} = \tilde{V}_i^{(\pi_i, \pi_{-i}^*)}$ for all $s \in S$. Thus, $\pi^*$ is a Nash policy as claimed. $\qquad \square$

An important property of the AlberDICE algorithm is that it maintains a sequence of factorizable joint policies, $\pi^t = \langle \pi_i^k \rangle_{i \in N}$, for any $t > 0$. Thus, after termination of the algorithm, the agents are guaranteed not only to reach an optimal state of the value function, but also an extracted joint policy that will be factorizable. This eliminates the problem of learning correlated policies during centralized training, which even if optimal, may not be useful during decentralized execution. This property of the AlberDICE is formally stated of Corollary 4.3. Its proof is immediate by the construction of the algorithm.

# E    Detailed Description of Practical Algorithm

## E.1    Numerically Stable Optimization

The practical issue in optimizing (14) is that it is unstable due to its inclusion of $\exp(\cdot)$, often causing exploding gradient problems:

$$L(\nu_i) := \min_{\nu_i} \bar{\rho}_i \alpha \hat{\mathbf{E}}_{x \in D_{\rho_i}} \left[ \exp \left( \tfrac{1}{\alpha} \hat{e}_{\nu_i}(s, a_i, \mathbf{a}_{-i}, s') - 1 \right) \right] + (1 - \gamma) \mathbb{E}_{s_0 \sim p_0} [\nu_i(s_0)]. \qquad (14)$$

where $\hat{e}_{\nu_i}(s, a_i, \mathbf{a}_{-i}, s') := r(s, a_i, \mathbf{a}_{-i}) - \alpha \log \frac{\pi_{-i}(\mathbf{a}_{-i}|s)}{\pi_{-i}^{D}(\mathbf{a}_{-i}|s, a_i)} + \gamma \nu_i(s') - \nu_i(s)$. To address this issue, we use the following alternative, which can be optimized stably.

$$\tilde{\mathcal{L}}(\tilde{\nu}_i) := \min_{\tilde{\nu}_i} \alpha \log \bar{\rho}_i \hat{\mathbf{E}}_{x \sim D_{\rho_i}} \left[ \exp \left( \tfrac{1}{\alpha} \hat{e}_{\tilde{\nu}_i}(s, a_i, \mathbf{a}_{-i}, s') \right) \right] + (1 - \gamma) \mathbb{E}_{s_0 \sim p_0} [\tilde{\nu}_i(s_0)] \qquad (31)$$

Note that the gradient $\nabla_x \log \hat{\mathbf{E}}_x[\exp(h(x))]$ is given by $\hat{\mathbf{E}}_x \left[ \frac{\exp(h(x))}{\hat{\mathbf{E}}_{x'}[\exp(h(x'))]} \nabla_x h(x) \right]$, which normalizes the value of $\exp(h(x))$ and thus it is numerically stable by preventing the exploding gradient issue. At first glance, it seems that optimizing (31) can result in a completely different solution to the solution of (14). However, as we will show in the followings, their optimal objective function values are the same and their optimal solutions only differ in a constant shift.

**Proposition E.1.** *Let $V^* = \arg\min_{\nu_i} L(\nu_i)$ and $\widetilde{V}^* = \arg\min_{\tilde{\nu}_i} \tilde{\mathcal{L}}(\tilde{\nu}_i)$ be the sets of optimal solutions of (14) and (31). Then, $L(\nu_i^*) = \tilde{\mathcal{L}}(\tilde{\nu}_i^*)$ holds for any $\nu_i^* \in V^*$ and $\tilde{\nu}_i^* \in \widetilde{V}^*$. Also, for any $\nu_i^* \in V$ and any $C \in \mathbb{R}$, $v_i^* + C \in \widetilde{V}^*$.*

We follow the proof steps in DemoDICE [9]. First, note that for any constant $C$, the advantage for $\nu_i + C$ is:

$$\hat{e}_{\nu_i+C}(s, a_i, \mathbf{a}_{-i}, s') = r(s, a_i, \mathbf{a}_{-i}) - \alpha \log \frac{\pi_{-i}(\mathbf{a}_{-i}|s)}{\pi_{-i}^{D}(\mathbf{a}_{-i}|s, a_i)} + \gamma(\nu_i(s') + C) - (\nu_i(s) + C)$$

$$= \hat{e}_{\nu_i}(s, a_i, \mathbf{a}_{-i}, s') - (1 - \gamma)C \qquad (32)$$

**Lemma E.2.** *For an arbitrary function $\nu_i$ and any constant $C$, the following equality holds,*

$$\tilde{\mathcal{L}}(\nu_i) = \tilde{\mathcal{L}}(\nu_i + C).$$

*Proof.* From the definition of $\tilde{\mathcal{L}}(\nu_i)$,

$$\tilde{\mathcal{L}}(\nu_i + C)$$

$$= (1 - \gamma)\mathbb{E}_{s_0 \sim p_0}[\nu_i(s_0) + C] + \alpha \log \bar{\rho}_i \hat{\mathbf{E}}_{x \sim D_{\rho_i}} \left[ \exp \left( \tfrac{1}{\alpha} \hat{e}_{\nu_i+C}(s, a_i, \mathbf{a}_{-i}, s') \right) \right]$$

$$= (1 - \gamma)\mathbb{E}_{s_0 \sim p_0}[\nu_i(s_0) + C] + \alpha \log \bar{\rho}_i \hat{\mathbf{E}}_{x \sim D_{\rho_i}} \left[ \exp \left( \tfrac{1}{\alpha} \hat{e}_{\nu_i}(s, a_i, \mathbf{a}_{-i}, s') - \tfrac{1}{\alpha}(1 - \gamma)C \right) \right] \text{ (by (32))}$$

$$= (1 - \gamma)\mathbb{E}_{s_0 \sim p_0}[\nu_i(s_0) + C] + \alpha \log \left\{ \exp \left( - \tfrac{1}{\alpha}(1 - \gamma)C \right) \bar{\rho}_i \hat{\mathbf{E}}_{x \sim D_{\rho_i}} \left[ \exp \left( \tfrac{1}{\alpha} \hat{e}_{\nu_i}(s, a_i, \mathbf{a}_{-i}, s') \right) \right] \right\}$$

$$= (1 - \gamma)\mathbb{E}_{s_0 \sim p_0}[\nu_i(s_0) + C] - (1 - \gamma)C + \alpha \log \bar{\rho}_i \hat{\mathbf{E}}_{x \sim D_{\rho_i}} \left[ \exp \left( \tfrac{1}{\alpha} \hat{e}_{\nu_i}(s, a_i, \mathbf{a}_{-i}, s') \right) \right]$$

$$= (1 - \gamma)\mathbb{E}_{s_0 \sim p_0}[\nu_i(s_0)] + \alpha \log \bar{\rho}_i \hat{\mathbf{E}}_{x \sim D_{\rho_i}} \left[ \exp \left( \tfrac{1}{\alpha} \hat{e}_{\nu_i}(s, a_i, \mathbf{a}_{-i}, s') \right) \right]$$

$$= \tilde{\mathcal{L}}(\nu_i)$$

$\square$

**Lemma E.3.** *For any function $\nu_i$, the following inequality always holds:*

$$L(\nu_i) \geq \tilde{\mathcal{L}}(\nu_i).$$

*Equality holds if and only if*

$$\bar{\rho}_i \hat{\mathbf{E}}_{x \sim D_{\rho_i}} \left[ \exp \left( \tfrac{1}{\alpha} \hat{e}_{\nu_i}(s, a_i, \mathbf{a}_{-i}, s') \right) \right] = 1$$

*Proof.* For any $y \geq 0$

$$y - 1 \geq \log y$$

and equality holds if and only if $y = 1$. Thus,

$$\bar{\rho}\hat{\mathbf{E}}_{x \sim D_{\rho_i}}\left[\exp\left(\tfrac{1}{\alpha}\hat{e}_{\nu_i}(s, a_i, \mathbf{a}_{-i}, s')\right)\right] - 1 \geq \log \bar{\rho}_i\hat{\mathbf{E}}_{x \sim D_{\rho_i}}\left[\exp\left(\tfrac{1}{\alpha}\hat{e}_{\nu_i}(s, a_i, \mathbf{a}_{-i}, s')\right)\right]$$

and equality holds if and only if

$$\bar{\rho}_i\hat{\mathbf{E}}_{x \sim D_{\rho_i}}\left[\exp\left(\tfrac{1}{\alpha}\hat{e}_{\nu_i}(s, a_i, \mathbf{a}_{-i}, s')\right)\right] = 1.$$

Finally, we obtain the following results:

$$\begin{aligned}
\tilde{L}(\nu_i) &= (1-\gamma)\mathbb{E}_{s_0 \sim p_0}[\nu_i(s_0)] + \bar{\rho}_i\alpha\hat{\mathbf{E}}_{x \in D_{\rho_i}}\left[\exp\left(\tfrac{1}{\alpha}\hat{e}_{\nu_i}(s, a_i, \mathbf{a}_{-i}, s') - 1\right)\right] \\
&\geq (1-\gamma)\mathbb{E}_{s_0 \sim p_0}[\nu_i(s_0)] + \alpha\log\left\{\bar{\rho}_i\hat{\mathbf{E}}_{x \in D_{\rho_i}}\left[\exp\left(\tfrac{1}{\alpha}\hat{e}_{\nu_i}(s, a_i, \mathbf{a}_{-i}, s') - 1\right)\right]\right\} + 1 \\
&\geq (1-\gamma)\mathbb{E}_{s_0 \sim p_0}[\nu_i(s_0)] + \alpha\log\left\{\bar{\rho}_i\hat{\mathbf{E}}_{x \in D_{\rho_i}}\left[\exp\left(\tfrac{1}{\alpha}\hat{e}_{\nu_i}(s, a_i, \mathbf{a}_{-i}, s') - 1\right)\right]\exp(1)\right\} \\
&= (1-\gamma)\mathbb{E}_{s_0 \sim p_0}[\nu_i(s_0)] + \alpha\log\left\{\bar{\rho}_i\hat{\mathbf{E}}_{x \in D_{\rho_i}}\left[\exp\left(\tfrac{1}{\alpha}\hat{e}_{\nu_i}(s, a_i, \mathbf{a}_{-i}, s')\right)\right]\right\} \\
&=: \tilde{\mathcal{L}}(\nu_i)
\end{aligned}$$

$\square$

**Lemma E.4.** *For any optimal solution $\tilde{\nu}_i^* = \arg\min_{\nu_i} \tilde{\mathcal{L}}(\nu_i)$, there is a constant $C$ such that $\tilde{\nu}_i^* + C$ is an optimal solution of $\min_{\nu_i} L(\nu_i)$.*

*Proof.* Let $\tilde{\nu}_i^*$ be an optimal solution of $\arg\min_{\nu_i} \tilde{\mathcal{L}}(\nu_i)$ and

$$C^* := \tfrac{\alpha}{1-\gamma}\log\bar{\rho}_i\hat{\mathbf{E}}_{x \sim D_{\rho_i}}\left[\exp\left(\tfrac{1}{\alpha}\hat{e}_{\nu_i}(s, a_i, \mathbf{a}_{-i}, s')\right)\right]$$

Then, $\hat{\nu}_i := \tilde{\nu}_i^* + C^*$ satisfies

$$\begin{aligned}
&\bar{\rho}_i\hat{\mathbf{E}}_{x \sim D_{\rho_i}}\left[\exp\left(\tfrac{1}{\alpha}\hat{e}_{\hat{\nu}_i}(s, a_i, \mathbf{a}_{-i}, s')\right)\right] \\
&= \bar{\rho}_i\hat{\mathbf{E}}_{x \sim D_{\rho_i}}\left[\exp\left(\tfrac{1}{\alpha}\hat{e}_{\tilde{\nu}_i^* + C^*}(s, a_i, \mathbf{a}_{-i}, s')\right)\right] \\
&= \bar{\rho}_i\hat{\mathbf{E}}_{x \sim D_{\rho_i}}\left[\exp\left(\tfrac{1}{\alpha}\hat{e}_{\tilde{\nu}_i^*}(s, a_i, \mathbf{a}_{-i}, s') - \tfrac{1-\gamma}{\alpha}C^*\right)\right] \\
&= \bar{\rho}_i\hat{\mathbf{E}}_{x \sim D_{\rho_i}}\left[\exp\left(\tfrac{1}{\alpha}\hat{e}_{\tilde{\nu}_i^*}(s, a_i, \mathbf{a}_{-i}, s')\right)\exp\left(-\tfrac{1-\gamma}{\alpha}C^*\right)\right] \\
&= 1.
\end{aligned}$$

Furthermore, $\hat{\nu}_i$ is also an optimal solution of $\min_{\nu_i} \tilde{\mathcal{L}}(\nu_i)$ by Lemma E.2 . Then, by the equality condition in Lemma E.3 ,

$$\tilde{L}(\hat{\nu}_i) = \tilde{\mathcal{L}}(\hat{\nu}_i) = \min_{\nu_i}\tilde{\mathcal{L}}(\nu_i) \leq \min_{\nu_i}\tilde{L}(\nu_i).$$

Thus, $\hat{\nu}_i$ is an optimal solution of $\min_{\nu_i} L(\nu_i)$. $\square$

**Lemma E.5.** *An optimal solution $\nu_i^* = \arg\min_{\nu_i} L(\nu_i)$ is also an optimal solution of $\min_{\nu_i} \tilde{\mathcal{L}}(\nu_i)$*

*Proof.* From Lemma E.3 ,

$$\min_{\nu_i} L(\nu_i) = L(\nu_i^*) \geq \tilde{\mathcal{L}}(\nu_i^*).$$

From Lemma E.4, $\min_{\nu_i} L(\nu_i)$ and $\min_{\nu_i} \tilde{\mathcal{L}}(\nu_i)$ have the same minimum value, and thus,

$$\tilde{\mathcal{L}}(\nu_i^*) \leq L(\nu_i^*) = \min_{\nu_i} L(\nu_i) = \min_{\nu_i} \tilde{\mathcal{L}}(\nu_i)$$

holds. $\square$

The aforementioned Proposition E.1 can now be proved.

**Proposition E.1.** *Let* $V^* = \arg\min_{\nu_i} L(\nu_i)$ *and* $\widetilde{V}^* = \arg\min_{\tilde{\nu}_i} \tilde{\mathcal{L}}(\tilde{\nu}_i)$ *be the sets of optimal solutions of (14) and (31). Then,* $L(\nu_i^*) = \tilde{\mathcal{L}}(\tilde{\nu}_i^*)$ *holds for any* $\nu_i^* \in V^*$ *and* $\tilde{\nu}_i^* \in \widetilde{V}^*$. *Also, for any* $\nu_i^* \in V$ *and any* $C \in \mathbb{R}$, $v_i^* + C \in \widetilde{V}^*$.

*Proof.* This holds from combining Lemma E.4 and Lemma E.5 □

### E.2 Policy Extraction

Finally, our practical AlberDICE optimizes (31) that yields $\tilde{\nu}_i^*$. However, $\tilde{\nu}_i^*$ itself is not a directly executable policy, so we should extract a policy from it. To this end, first note that the optimal policy $\pi_i^*$ is encoded in $w_i^*$ as a form of stationary distribution correction ratios:

$$w_i^*(s, a) = \frac{d^{\pi_i^*}(s, a_i)}{d^D(s, a_i)} \quad \text{(optimal solution of Eq. (8))} \tag{33}$$

Then, $w_i^*(s, a)$ can be represented in terms of $\tilde{\nu}_i^*$ as follows:

$$w_i^*(s, a) = \exp\left(\tfrac{1}{\alpha} e_{\nu_i^*}(s, a_i) - 1\right) \quad \text{(Eq. (22))} \tag{34}$$

$$\propto \exp\left(\tfrac{1}{\alpha} e_{\tilde{\nu}_i^*}(s, a_i)\right) \quad \text{(by Proposition E.1)} \tag{35}$$

where $e_{\tilde{\nu}_i^*}(s, a_i) = \mathbb{E}_{\substack{\mathbf{a}_{-i} \sim \boldsymbol{\pi}_{-i}(s) \\ s' \sim P(s, a_i, \mathbf{a}_{-i})}} \left[\hat{e}_{\tilde{\nu}_i^*}(s, a_i, \mathbf{a}_{-i}, s')\right]$ and $\hat{e}_{\tilde{\nu}_i^*}(s, a_i, \mathbf{a}_{-i}, s') = r(s, a_i, \mathbf{a}_{-i}) - \alpha \log \frac{\boldsymbol{\pi}_{-i}(\mathbf{a}_{-i}|s)}{\boldsymbol{\pi}_{-i}^D(\mathbf{a}_{-i}|s, a_i)} + \gamma \tilde{\nu}_i^*(s') - \tilde{\nu}_i^*(s)$. Finally, we extract a policy from $w_i^*$ via I-projection policy extraction method introduced in equation (15).

$$\min_{\pi_i} D_{\text{KL}}\left(d^D(s)\pi_i(a_i|s)\boldsymbol{\pi}_{-i}(\mathbf{a}_{-i}|s) || d^D(s)\pi_i^*(a_i|s)\boldsymbol{\pi}_{-i}(\mathbf{a}_{-i}|s)\right) \tag{36}$$

$$= \hat{\mathbf{E}}_{s \in D, a_i \sim \pi_i, \mathbf{a}_{-i} \sim \boldsymbol{\pi}_{-i}(\mathbf{a}_{-i}|s)}\left[\log \frac{d^D(s)\pi_i(a_i|s)\boldsymbol{\pi}_{-i}(\mathbf{a}_{-i}|s)}{d^D(s)\pi_i^*(a_i|s)\boldsymbol{\pi}_{-i}(\mathbf{a}_{-i}|s)}\right] \tag{37}$$

$$= \hat{\mathbf{E}}_{s \sim D, a_i \sim \pi_i}\left[\log \frac{d^D(s)\pi_i^D(a_i|s)}{d^{\pi_i^*}(s)\pi_i^*(a_i|s)} + \log \frac{\pi_i(a_i|s)}{\pi_i^D(a_i|s)} + \underbrace{\log \frac{d^{\pi_i^*}(s)}{d^D(s)}}_{\text{constant for } \pi_i}\right] \tag{38}$$

$$= \hat{\mathbf{E}}_{s \in D, a_i \sim \pi_i}\left[\log \frac{d^D(s, a_i)}{d^{\pi_i^*}(s, a)} + D_{\text{KL}}\left(\pi_i(a_i|s) || \pi_i^D(a_i|s)\right)\right] + C_1 \tag{39}$$

$$= \hat{\mathbf{E}}_{s \in D, a_i \sim \pi_i}\left[-\log w_i^*(s, a_i) + D_{\text{KL}}\left(\pi_i(a_i|s) || \pi_i^D(a_i|s)\right)\right] + C_1 \tag{40}$$

$$= \hat{\mathbf{E}}_{s \in D, a_i \sim \pi_i}\left[-\tfrac{1}{\alpha} e_{\tilde{\nu}_i^*}(s, a_i) + D_{\text{KL}}\left(\pi_i(a_i|s) || \pi_i^D(a_i|s)\right)\right] + C_1 + C_2 \quad \text{(by (35))} \tag{41}$$

where $C_1$ and $C_2$ denote some constants. $\pi_i^D(a_i|s)$ is a data policy for $i$-th agent, which is pretrained by maximizing the log-likelihood:

$$\max_{\pi_i^D} \hat{\mathbf{E}}_{(s, a_i) \in D}\left[\log \pi_i^D(a_i|s)\right] \tag{42}$$

Eq. (41) can be understood as KL-regularized policy optimization, where we aim to maximize $e_{\tilde{\nu}_i^*}(s, a_i)$ (analogous to critic value) while not deviating too much from the data policy, whose trade-off is controlled by the hyperparameter $\alpha$. Finally, to enable $e_{\tilde{\nu}_i^*}$ to be evaluated at every action $a_i$, we train an additional parametric function $e_i$ (implemented as an MLP that takes $(s, a_i)$ as an input and outputs a scalar value) by minimizing the mean squared error with a conservative regularization term $\mathcal{R}(e_i)$ introduced in CQL [11] to penalize OOD action values:

$$\min_{e_i} \hat{\mathbf{E}}_{(s, a_i, \mathbf{a}_{-i}, s') \in D_{\rho_i}}\left[\left(e_i(s, a_i) - \hat{e}_{\tilde{\nu}_i^*}(s, a_i, \mathbf{a}_{-i}, s')\right)^2\right] + \mathcal{R}_{\text{CQL}}(e_i) \tag{43}$$

where $\mathcal{R}_{\text{CQL}}(e_i) := \alpha_{\text{CQL}}\hat{\mathbf{E}}_{(s, a_i) \sim D_{\rho_i}}\left[\log \sum_{a_i} \exp\left(e_i(s, a_i)\right) - \mathbb{E}_{a_i \sim \pi_i^D}[e_i(s, a_i)]\right]$. We used $\alpha_{\text{CQL}} = 0.1$ for all experiments.

### E.3 Pseudocode of AlberDICE

To sum up, AlberDICE computes the best response of agent $i$ by optimizing $\nu_i$, which corresponds to obtaining a stationary distribution correction ratios of the optimal policy. Then, we extract a policy by training $e$-network and performing I-projection as described in Section E.2.

We assume $\pi_i^D$, $\boldsymbol{\pi}_{-i}^D$, $\nu_i$, $e_i$, and $\pi_i$ are parameterized by $\beta_i$, $\beta_{-i}$, $\theta_i$, $\psi_i$, and $\phi_i$, respectively[10]. Then, we optimize the parameters via stochastic gradient descent (SGD). The entire loss functions to optimize the parameters are summarized in the following:

$$J(\beta_i) := -\hat{\mathbf{E}}_{(s,a_i)\in D}\big[\log \pi_{\beta_i}^D(a_i|s)\big] \tag{44}$$

$$J(\beta_{-i}) := -\hat{\mathbf{E}}_{(s,a_i,\mathbf{a}_{-i})\in D}\Big[\sum_{j=1,j\neq i}^N \log \pi_{\beta_{-i}}^D(a_j|s,a_i,a_{<j})\Big] \tag{45}$$

$$J(\theta_i) := \alpha \log \bar{\rho}_i \hat{\mathbf{E}}_{x\sim D_{\rho_i}}\Big[\exp\Big(\tfrac{1}{\alpha}\hat{e}_{\nu_{\theta_i}}(s,a_i,\mathbf{a}_{-i},s')\Big)\Big] + (1-\gamma)\mathbb{E}_{s_0\sim p_0}[\nu_{\theta_i}(s_0)] \tag{46}$$

$$J(\psi_i) := \hat{\mathbf{E}}_{(s,a_i,\mathbf{a}_{-i},s')\in D_{\rho_i}}\Big[\big(e_{\psi_i}(s,a_i) - \hat{e}_{\tilde{\nu}_{\theta_i}}(s,a_i,\mathbf{a}_{-i},s')\big)^2\Big] \tag{47}$$
$$+ \alpha_{\text{CQL}}\hat{\mathbf{E}}_{s\in D_{\rho_i}}\Big[\big(\log\sum_{a_i}\exp\big(e_{\psi_i}(s,a_i)\big) - \mathbb{E}_{a_i\sim\pi_{\beta_i}^D}[e_{\psi_i}(s,a_i)]\big)\Big]$$

$$J(\phi_i) := \hat{\mathbf{E}}_{s\in D}\Big[\sum_{a_i}\pi_{\phi_i}(a_i|s)\big(-e_{\psi_i}(s,a_i) + \alpha\log\tfrac{\pi_{\phi_i}(a_i|s)}{\pi_{\beta_i}^D(a_i|s)}\big)\Big] \tag{48}$$

The pseudocode of AlberDICE is presented in Algorithm 1.

---

**Algorithm 1** AlberDICE

---

**Input:** A dataset $D := \{(s,\mathbf{a},r,s')_k\}_{k=1}^{|D|}$, a set of initial states $D_0 := \{s_{0,k}\}_{k=1}^{|D_0|}$, data policy networks $\{(\pi_{\beta_i}^D, \boldsymbol{\pi}_{\beta_{-i}}^D)\}_{i=1}^N$ with parameters $\{(\beta_i, \beta_{-i})\}_{i=1}^N$, $\nu$-networks $\{\nu_{\theta_i}\}_{i=1}^N$ with parameters $\{\theta_i\}_{i=1}^N$, $e$-networks $\{e_{\psi_i}\}_{i=1}^N$ with parameters $\{\psi_i\}_{i=1}^N$, policy networks $\{\pi_i^\phi\}_{i=1}^N$ with parameters $\{\phi_i\}_{i=1}^N$, and a learning rate $\eta$

1: Pretrain (auto-regressive) data policies $\big\{\big(\pi_{\beta_i}^D(a_i|s), \boldsymbol{\pi}_{\beta_{-i}}^D(\mathbf{a}_{-i}|s,a_i)\big)\big\}_{i=1}^N$ by minimizing (44-45).
2: **for** each iteration until convergence **do**
3:    **for** each agent $i\in\mathcal{N}$ **do**
4:       Sample mini-batches from $s_0\sim D_0$ and $x\sim D$.
5:       Compute the importance ratio $\rho_i(x) = \frac{\prod_{j\neq i}\pi_{\phi_j}(a_j|s)}{\boldsymbol{\pi}_{\beta_{-i}}^D(\mathbf{a}_{-i}|s,a_i)}$ for each sample $x$.   (Eq. (11))
6:       Perform resampling with probability proportional to $\rho_i(x)$, which constitutes the resampled dataset $D_{\rho_i}$.
7:       Perform SGD updates using $D_0$ and $D_{\rho_i}$:
$$\theta_i \leftarrow \theta_i - \eta\nabla_{\theta_i}J(\theta_i) \quad (\text{Eq. (46)})$$
$$\psi_i \leftarrow \psi_i - \eta\nabla_{\psi_i}J(\psi_i) \quad (\text{Eq. (47)})$$
$$\phi_i \leftarrow \phi_i - \eta\nabla_{\phi_i}J(\psi_i) \quad (\text{Eq. (48)})$$
8:    **end for**
9: **end for**
**Output:** Factorized policies $\{\pi_{\phi_i}(a_i|s)\}_{i=1}^N$

---

---

[10]To increase scalability, we use shared parameters and an additional agent ID input to train $\beta_i$ for $\pi_i^D$ in all experiments.

# F Dataset Details

## F.1 Bridge

The *optimal* dataset (500 trajectories) was constructed by a hand-crafted (multi-modal) optimal policy which randomizes between Agent 1 crossing the bridge first while Agent 2 retreats, and vice-versa. The *mix* dataset is a mixture between 500 trajectories from the *optimal* dataset and 500 trajectories generated by a uniform random policy.

## F.2 Multi-Robot Warehouse (RWARE)

For the data collection policy used to construct the dataset, we train Multi-Agent Transformers (MAT) [37] which takes an autoregressive policy structure and thus is able to generate diverse behavior. We further train MAT over 3 random seeds, and generate a *expert* dataset with a mixture of diverse behaviors.

## F.3 Google Research Football

Similar to the dataset collection procedure in RWARE, we use MAT to generate a *medium-expert* dataset in order to ensure that agents score goals in different ways. Similar to Bridge, we construct a dataset of 2000 trajectories where 1000 trajectories have medium performance (roughly 60% performance of the expert policies) and another 1000 from fully trained "expert" MAT policies.

## F.4 SMAC

We use the public dataset provided by [31].

# G Matrix Game Results

In Table 6, we show results for the converged policies in the Matrix Game presented in Section 5 of the main text and shown again in Figure 6.

|   | $A$ | $B$ |
|---|---|---|
| $A$ | 0 | 1 |
| $B$ | 1 | −2 |

Figure 6: XOR Game with Penalty

As expected, all algorithms converge to the optimal action $AB$ for the dataset $D_{(a)}$. However, AlberDICE is the only algorithm which can choose the optimal action deterministically for all 4 datasets, showing robustness even when the environment has multiple global optima and the dataset is generated by a mixture of diverse policies. Here we can consider any dataset containing both $AB$ and $BA$ as a mixture of diverse data collection policies where the two agents cooperate to select the optimal actions but in different ways. For OMAR, we see that it is able to learn the optimal policy for $D_{(b)}$. However, as discussed in the Introduction of the main text, OMAR degenerates to the sub-optimal action $BB$ for dataset $D_{(c)}$ because each agent acts independently and assumes the other agent chooses the individual action $A$ with a $\frac{2}{3}$ probability. Finally, we also note that BC can fail and converge to OOD joint actions even in $D_{(b)}$ where the dataset is optimal.

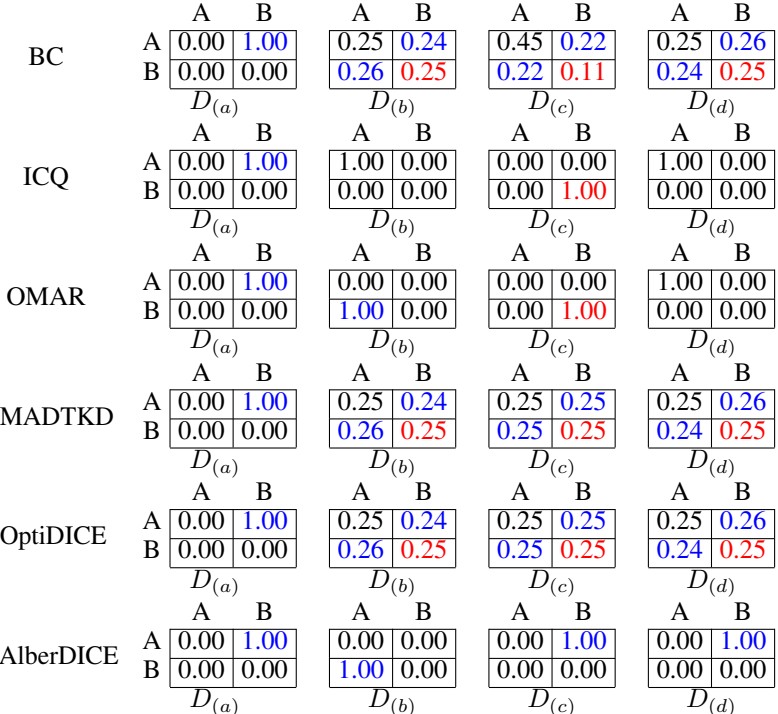

Table 6: Policy values after convergence for the Matrix Game in Figure 2. The policy values are calculated by multiplying the individual policy values for each agent i.e. $\pi = \pi_1 \times \pi_2$. The datasets consist of $D_{(a)} = \{AB\}, D_{(b)} = \{AB, BA\}, D_{(c)} = \{AA, AB, BA\}, D_{(d)} = \{AA, AB, BA, BB\}$.

# H  Bridge Policy Visualizations

The visualizations for learned policies of AlberDICE, OptiDICE, ICQ, OMAR, BC, and MADTKD are shown for all state possibilities. From Figure 7, it is clear that AlberDICE is the only algorithm which reliably chooses a deterministic action (Left, Left) at the initial state. However, since the visualizations are provided for the optimal dataset which has a small coverage of states, the policy values may still be sub-optimal at OOD states. The results for OMAR in Figure 10 shows that the agents are acting independently without regard for the other agents, and converges to (Left, Right) at the initial state which results in a collision. Finally, we note that the visualizations for MADTKD may not exactly correspond to the policy values used in the experimental results in Section 5. This is because MADTKD uses a history-dependent Transformer policy and it is not clear how to visualize the policy values depending on the history. The visualizations shown in Figure 11 assume that each agent is at the given state in the initial timestep, which is different from encountering that state after many timesteps. Nonetheless, the results for the true initial state shown in the central portions of the figure are consistent with the quantitative results in Section 5.

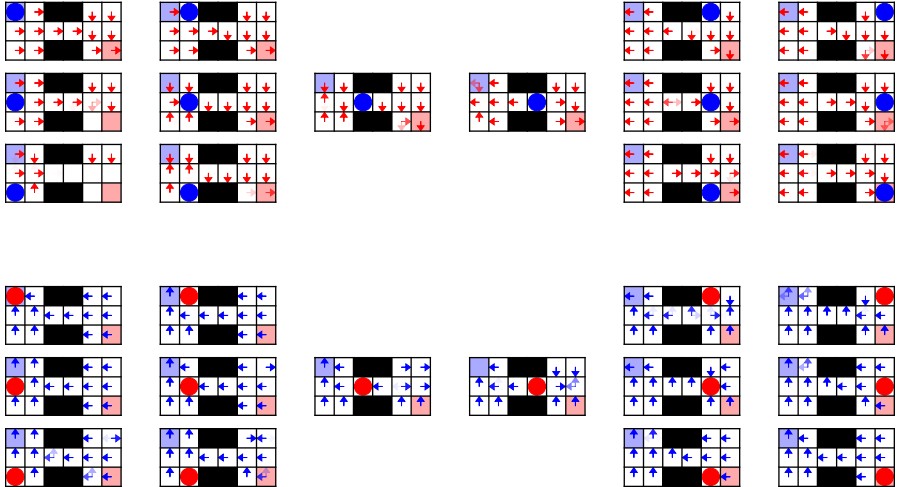

Figure 7: Policy Visualizations for **AlberDICE** on the Bridge (Hard) environment for the optimal dataset where the arrows show the probability of choosing a particular action given that the other agents are in ● and ● for agents 1 and 2, respectively.

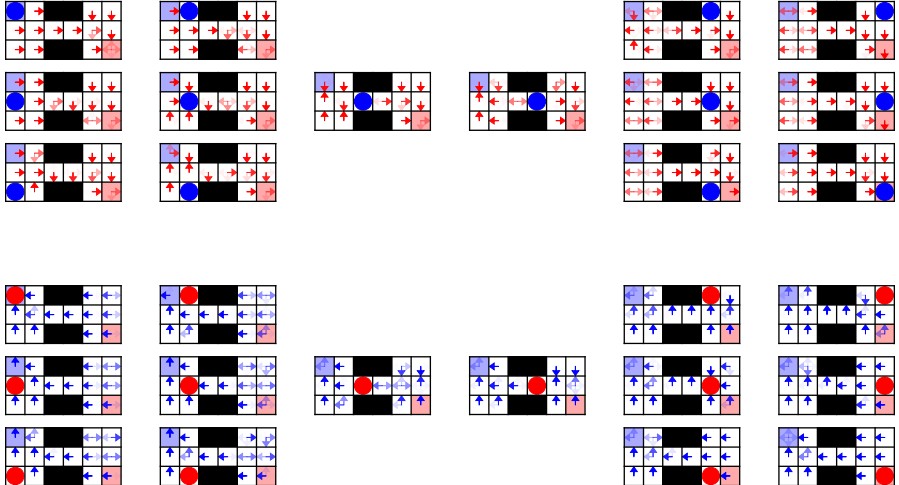

Figure 8: Policy Visualizations for **OptiDICE** on the Bridge (Hard) environment for the optimal dataset where the arrows show the probability of choosing a particular action given that the other agents are in ● and ● for agents 1 and 2, respectively.

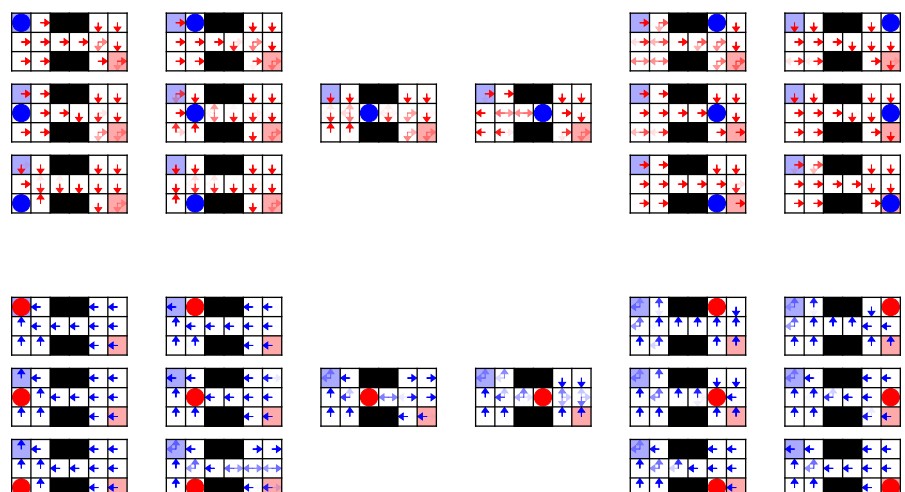

Figure 9: Policy Visualizations for **ICQ** on the Bridge (Hard) environment for the optimal dataset where the arrows show the probability of choosing a particular action given that the other agents are in ● and ● for agents 1 and 2, respectively.

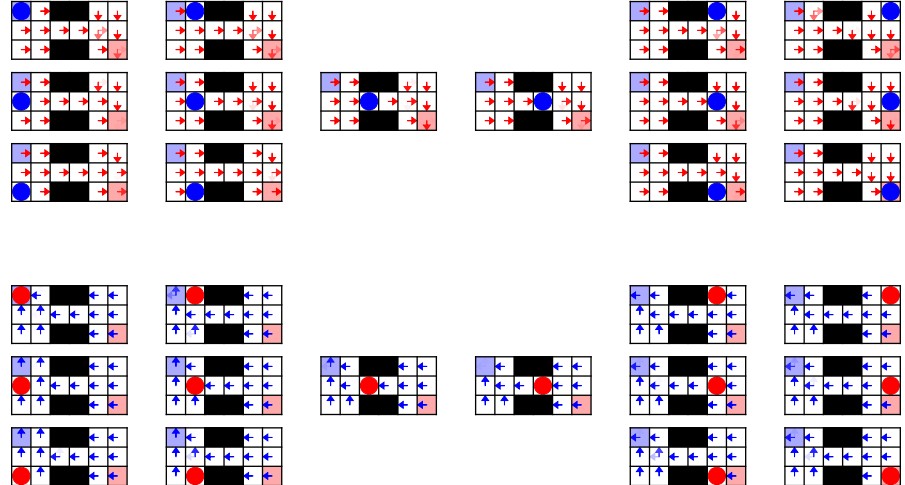

Figure 10: Policy Visualizations for **OMAR** on the Bridge (Hard) environment for the optimal dataset where the arrows show the probability of choosing a particular action given that the other agents are in ● and ● for agents 1 and 2, respectively.

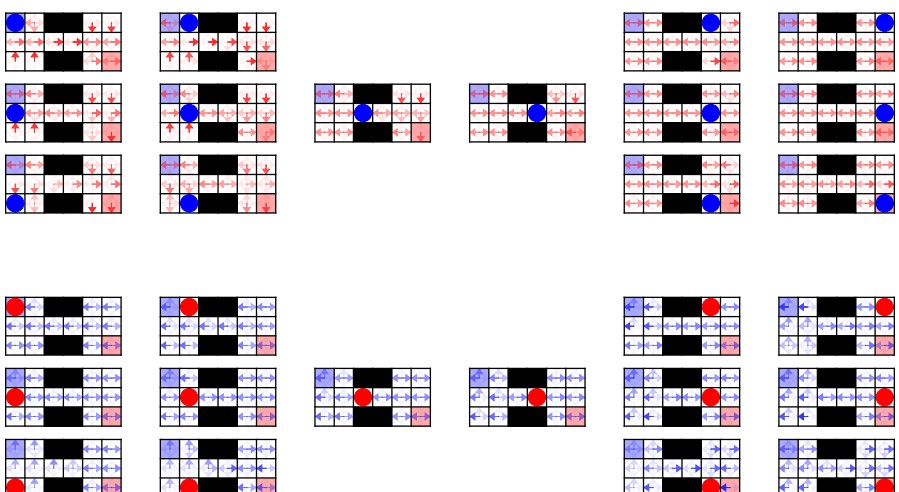

Figure 11: Policy Visualizations for **MADTKD** on the Bridge (Hard) environment for the optimal dataset where the arrows show the probability of choosing a particular action given that the other agents are in ● and ● for agents 1 and 2, respectively.

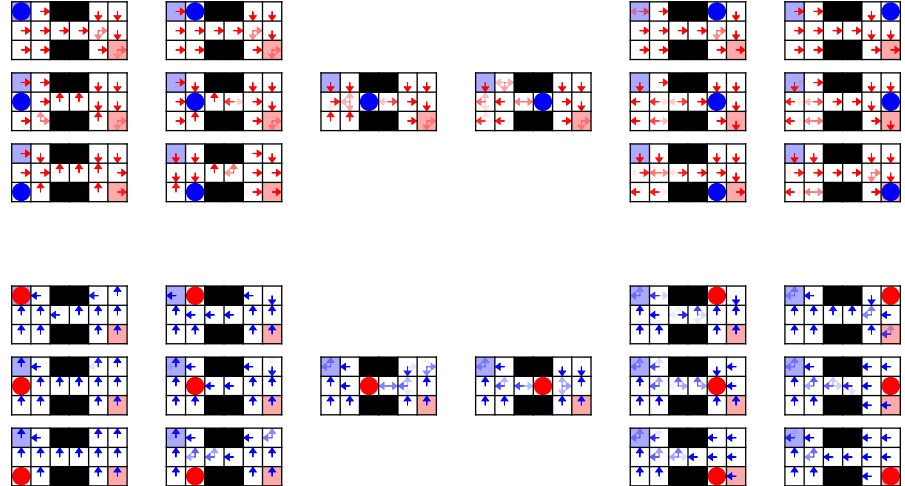

Figure 12: Policy Visualizations for **BC** on the Bridge (Hard) environment for the optimal dataset where the arrows show the probability of choosing a particular action given that the other agents are in ● and ● for agents 1 and 2, respectively.

# I   Additional Ablation Results

Here we show ablation results for hyperparameter $\alpha$ which controls the degree of conservatism. Table 7 shows that AlberDICE is not too sensitive to the hyperparameter $\alpha$ as long as it is within a reasonable range.

| $\alpha$ | 0.001 | 0.01 | 0.1 | 1 | 10 | 100 | 1000 |
|---|---|---|---|---|---|---|---|
| CA-Hard (GRF) ($N = 4$) | $0.00 \pm 0.00$ | $0.08 \pm 0.11$ | $0.82 \pm 0.03$ | $0.84 \pm 0.06$ | $0.78 \pm 0.09$ | $0.78 \pm 0.11$ | $0.79 \pm 0.11$ |
| Corridor (SMAC) ($N = 8$) | $0.00 \pm 0.00$ | $0.00 \pm 0.00$ | $0.92 \pm 0.02$ | $0.97 \pm 0.01$ | $0.95 \pm 0.03$ | $0.92 \pm 0.03$ | $0.91 \pm 0.04$ |

Table 7: Ablation Study for Conservatism Hyperparameter $\alpha$ (over 3 random seeds)

# J   Implementation Details

## J.1   AlberDICE for Dec-POMDP

While our paper focuses on MMDPs, our experimental results show that we can extend AlberDICE to Dec-POMDPs $G = \langle \mathcal{N}, \mathcal{S}, \mathcal{A}, r, P, p_0, \gamma, \Omega, O \rangle$ by using partial observations and a history-dependent policy in place of a state-dependent policy for each agent. In Dec-POMDP, each agent $i$ observes individual observations $o_i \in \Omega$ which is given by the observation function $O : \mathcal{S} \times \mathcal{A} \to \Omega$. Each agent makes decision based on the observation-action history $\tau_i \in (\Omega \times A)^{t-1} \times \Omega$, where each agent's (decentralized) policy is represented as $\pi_i(a_i|\tau_i)$.

For the Matrix Game, Bridge and GRF, we used an MLP policy since the partial observations for each agent correspond with the global state (i.e., each individual policy is conditioned only on its current observation $o_i$, rather than the entire history). For Warehouse, each individual policy uses the partial observations as input, which is the 3x3 neighborhood surrounding each agent. We utilize a Transformer-based policy in order for the agent to condition on the history of local observations and actions, while speeding up training in comparison to Recurrent Neural Networks (RNNs). This same Transformer-based policy is used for all baselines as well.

During centralized training, AlberDICE uses the global state $s$ for training $\boldsymbol{\pi}_{\beta_{-i}}^D(\mathbf{a}_{-i}|s, a_i), \nu_{\theta_i}(s)$, and $e_{\psi_i}(s, a)$ for all environments, where its training procedure is identical to the MMDP training procedure described in Appendix E.3. Only the policy extraction step is different for Dec-POMDP, where each agent's history-dependent policies are trained by:

$$\min_{\beta_i} -\hat{\mathbf{E}}_{(\tau_i, a_i) \in D} \left[ \log \pi_{\beta_i}^D(a_i|\tau_i) \right] \tag{49}$$

$$\min_{\phi_i} \hat{\mathbf{E}}_{(s, a_i, \tau_i) \in D} \left[ \sum_{a_i} \pi_{\phi_i}(a_i|\tau_i) \left( -e_{\psi_i}(s, a) + \alpha \log \frac{\pi_{\phi_i}(a_i|\tau_i)}{\pi_{\beta_i}^D(a_i|\tau_i)} \right) \right] \tag{50}$$

## J.2   Hyperparameter Details

We conduct minimal hyperparameter tuning for all algorithms for fair comparisons. It is also worth noting that in offline RL, it is important to develop algorithms which require minimal hyperparameter tuning [23]. We chose the best values for $\alpha$ for both AlberDICE and OptiDICE between $[0.1, 1, 5, 10]$ on N=2 (Tiny) for RWARE and RPS (2 agents) for GRF. The best values were then used for all scenarios thereafter. The final values used were $\alpha = 0.1$ (GRF) and $\alpha = 1$ (RWARE) for AlberDICE and $\alpha = 1$ (GRF, RWARE) for OptiDICE. We found that the performance gaps between different hyperparameters were minimal as long as they were within a reasonable range where training is numerically stable.

For ICQ [40], we found that the algorithm tends to become numerically unstable after a certain number of training epochs even with sufficient hyperparameter tuning due to the exploding Q values, especially in the GRF and SMAC environment.

# K   Computational Resources

For Warehouse experiments, we utilized a single NVIDIA Geforce RTX 3090 graphics processing unit (GPU). The experiments for running AlberDICE took 5H, 14H, and 29H for 2, 4, 6 agent environments, respectively.

