# OpenReview forum: "AlberDICE: Addressing Out-Of-Distribution Joint Actions in Offline Multi-Agent RL via Alternating Stationary Distribution Correction Estimation"
_NeurIPS.cc/2023/Conference — NeurIPS 2023 poster_

### Official Review · Reviewer_5Yz6 · 2023-07-03

**Soundness:** 2 fair
**Presentation:** 3 good
**Contribution:** 2 fair
**Rating:** 5
**Confidence:** 4

**Summary:**

the paper presents a novel approach, AlberDICE, an offline MARL algorithm that performs centralized training of individual agents based on stationary distribution optimization, to tackle the distribution shift challenge in offline MARL. The proposed method avoids OOD joint action selection by computing the best response of one agent at a time, thereby circumventing the exponential complexity of MARL. The authors provide theoretical proof of convergence to Nash policies and demonstrate through experiments that AlberDICE outperforms baseline algorithms on standard MARL benchmarks.



**Strengths:**

1.	It proposes a novel solution to address the distribution shift problem in offline reinforcement learning by performing centralized training of individual agents and optimizing the stationary distribution. This approach effectively mitigates the issue of excessive distribution shift in offline multi-agent reinforcement learning.
2.	The authors avoid the exponential complexity challenge in multi-agent reinforcement learning by computing the best response of individual agents in a sequential manner, reducing the computational burden.
3.	The paper provides theoretical proof of convergence to Nash policies through the alternating optimization process, offering a theoretical guarantee for the effectiveness of the algorithm.
4.	Experimental results demonstrate that AlberDICE outperforms baseline algorithms significantly on standard multi-agent reinforcement learning benchmarks, validating the performance and effectiveness of the algorithm.


**Weaknesses:**

1.	This paper considers the other agents’ policies are fixed and optimized based on the distribution correction technique. However, what if the data set contains multiple cooperation strategies?
2.	Even though there are many derivations in Section 3, many are derived from existing work [1-3]. Section 3 should be simplified to highlight the contribution of this work.
3.	This works adopt BC(Behavioral Cloning) to learn \pi^D_{-i}. When the quality of the data set is not high, such as the medium data set, the effect of BC is very poor. Will this affect the performance of the algorithm?
4.	There is little discussion on the limitations and applicability of the algorithm, and there is a lack of in-depth exploration of special cases or problems, which may restrict the generalization and application of the algorithm.
5.	Necessary ablation experiments are required, such as the robustness of parameters, the selection of network models, etc.



**Questions:**

Please refer to the weakness.

---

> ### Author Rebuttal · Authors · 2023-08-10
>
> # Response to Reviewer $\text{\textcolor{blue}{5Yz6}}$
>
>
> Thank you for taking the time to provide detailed feedback on our paper. Please feel free to let us know if any of our responses below need further clarification. We’d be more than happy to address them during the reviewer-author discussion period.
>
>
> > **W1. This paper considers the other agents’ policies are fixed and optimized based on the distribution correction technique. However, what if the data set contains multiple cooperation strategies?**
>
>
> Fixed other agents' policies refer to our optimizing target policy (which changes over iterations), not the data-collection policy for other agents. Our method is not limited to the case where the dataset was collected by a single agent. In the experiments, we are already using the dataset *containing multiple cooperative strategies*. For example, in RWARE and GRF, we used 3 different agents to construct medium/expert dataset respectively. For Bridge-mix, we are using the dataset, which is a mixture of expert and random agents' experiences. We have discussed the details about the datasets in Section A.6 of the Appendix. Finally, even with the dataset that contains multiple strategies, AlberDICE outperforms the baselines.
>
>
> > **W2. Even though there are many derivations in Section 3, many are derived from existing work [1-3]. Section 3 should be simplified to highlight the contribution of this work.**
>
>
> Although the derivation of AlberDICE shares similarity with OptiDICE, its derivation is not identical nor trivial. While it is true that fixing other agents results in a single-agent RL problem, in *offline multi-agent* RL, we should additionally consider the distribution shift of other agents' policy, which corresponds to the case where transition/reward dynamics change in the reduced single-agent problem. The additional challenge here is that we do not have any samples for the 'shifted' dynamics for the reduced single-agent MDP. Importance sampling can address such a distribution shift in principle, but it suffers from large variance especially when the number of agents is large. To alleviate the large variance issue, we adopt importance resampling (IR) and fully derive the objective function that can be optimized very stably, which is one of our main contributions of this work. We will simplify Section 3 to make these points clear in the final version of the paper.
>
> > **W3. This works adopt BC(Behavioral Cloning) to learn $\pi^D_{-i}$. When the quality of the data set is not high, such as the medium data set, the effect of BC is very poor. Will this affect the performance of the algorithm?**
>
> Every offline (MA)RL algorithm is affected by the quality of the dataset. Still, if the dataset coverage over state-action space is sufficiently large and diverse, we can obtain a well-performing policy by controlling the conservatism hyperparameter (e.g. $\alpha$ in AlberDICE). When we tested AlberDICE on a random dataset (very poor in terms of data-collection policy's performance) in Bridge, it still could converge to an optimal joint policy. We believe that there is no reason why AlberDICE should suffer much more adversely from bad dataset quality than other algorithms. Please also note that $\pi^D_{-i}$ is only used for importance resampling which is to correct the distribution shift between the policy being learned and the policy used for data collection. For more details about how the datasets were constructed, please refer to Section A.6 of the Appendix.
>
>
> > **W4. There is little discussion on the limitations and applicability of the algorithm, and there is a lack of in-depth exploration of special cases or problems, which may restrict the generalization and application of the algorithm.**
>
>
> We have provided a discussion on the limitations of our approach in Section A.10 in the Appendix regarding the use of Nash policies, and how it may not necessarily converge to the global optimum. For example, consider the following matrix game:
> ```
> ~ A B
> A 1 0
> B 0 2
> ```
> If AlberDICE starts its optimization from a deterministic target policy that selects (A,A), it may not be able to escape from the local optimum due to its nature of computing the best response. Still, AlberDICE starting from a randomly initialized policy works well in practice, yielding better-performing policies than the baselines across multiple domains. Finally, throughout the paper we have stayed consistent in our problem setting (i.e. common reward settings and Multi-Agent MDPs), which is commonly used in all previous offline MARL baselines [5-8]. Please let us know if there are any specific limitations and we would be happy to provide any further clarifications.
>
> > **W5. Necessary ablation experiments are required**
>
>
> We have added additional ablation results for the choice of hyperparameter $\alpha$ that controls conservatism in the PDF in the general response, which is also shown below:
>
> | $\alpha$ |     CA-Hard (GRF)    | Corridor (SMAC)|
> | :------------------:| :---------:| :---------:|
> | 0.001 | 0.00 ± 0.00| 0.00 ± 0.00 |
> | 0.01 | 0.08 ± 0.11 | 0.00 ± 0.00 |
> | 0.1 | 0.82 ± 0.03| 0.92 ± 0.02 |
> | 1 | 0.84 ± 0.06| 0.97 ± 0.01|
> | 10 | 0.78 ± 0.09|  0.95 ± 0.03 |
> | 100 | 0.78 ± 0.11 | 0.92 ± 0.03 |
> | 1000 | 0.79 ± 0.11| 0.91 ± 0.04 |
>
> Table B2: Ablation results for the conservatism hyperparameter $\alpha$. Mean success rate and standard error shown over 3 random seeds.
>
> Here the results show that AlberDICE is not too sensitive to the hyperparameter $\alpha$ as long as it is within a reasonable range.

---

> > ### Comment · Reviewer_5Yz6 · 2023-08-18
> >
> > Thank you for your response. It has addressed some of my concerns. However, given the current state of the manuscript, there are numerous areas that require revisions.

---

> > > ### Author Response · Authors · 2023-08-18
> > >
> > > Thank you for the response as well as for increasing our score. We appreciate the detailed review and we will incorporate the points raised into the final version of the paper. Please let us know again if there are further concerns which we can clarify.

---

### Official Review · Reviewer_aFwQ · 2023-07-06

**Soundness:** 2 fair
**Presentation:** 2 fair
**Contribution:** 2 fair
**Rating:** 5
**Confidence:** 3

**Summary:**

This paper proposes a new algorithm called AlberDICE for addressing the challenge of out-of-distribution joint actions in offline Multi-Agent Reinforcement Learning (MARL). The proposed algorithm performs centralized training of individual agents based on stationary distribution optimization, which circumvents the exponential complexity of MARL by computing the best response of one agent at a time while effectively avoiding OOD joint action selection. The authors show that the alternating optimization procedure converges to Nash policies theoretically and demonstrate that AlberDICE significantly outperforms baseline algorithms on a standard suite of MARL benchmarks in experiments.

**Strengths:**

- Nice writing flow for solving the original LP problem. The paper provides a clear and concise explanation of the proposed approach and its implementation.
- AlberDICE is shown to produce much stronger policies than baselines in experiments conducted in many multi-agent environments.

**Weaknesses:**

- Since $\pi_{-i}$ is fixed, the process in Section 3 is quite similar to that of OptiDICE, including Lagrangian formation, the closed-form solution, and I-projection used in policy extraction. Also, the distribution shift/OOD problem is solved by the series of DICE. It weakens the contributions of this paper.
- The key point for AlberDICE is Theorem 4.2 which NE can be achieved. However, it is theoretically unclear to me why after the optimization, the inequality between lines 591-592 in the appendix holds. Let's consider an extreme case in the Rock Paper Scissors game. If the dataset only contains draw cases, how does AlberDICE reach NE? Maybe Theorem 4.2 lacks the necessary assumptions.

minors/typos
- "+" before the second line of eq.5
- In line 188, "pretrained." -> "pretrained,"
- "=" before eq.16 -> "$min_{\pi_i}$"

**Questions:**

- $x$ in eq.10-14 is sample $(s, a, s')$, $e_v$ is also a function of $(s,a)$. How eq.14 is state-dependent instead of state-action-dependent?



**Limitations:**

The paper does not explicitly mention any limitations of the proposed method. The paper has no potential negative societal impact.

---

> ### Author Rebuttal · Authors · 2023-08-10
>
> # Response to Reviewer $\text{\textcolor{red}{aFwQ}}$
>
> Thank you very much for the detailed review. Please feel free to let us know if any of our responses below need further clarification. We’d be more than happy to address them during the reviewer-author discussion period.
>
>
> > **W1. Since $\pi_{-i}$ is fixed, the process in Section 3 is quite similar to that of OptiDICE, including Lagrangian formation, the closed-form solution, and I-projection used in policy extraction. Also, the distribution shift/OOD problem is solved by the series of DICE. It weakens the contributions of this paper.**
>
> To the best of our knowledge, AlberDICE is the first offline multi-agent RL algorithm that precisely addresses the issue of selecting **joint** out-of-distribution action in a principled manner. While it is true that fixing other agents results in a single-agent RL problem, solving the induced single-agent problem in an "offline" manner is NOT trivial. Note that other agents' policy updates entail changes in the transition/reward dynamics of the (reduced) single-agent problem, but we do not have any samples for the 'shifted' dynamics in each reduced single-agent MDP. This is particularly problematic when applying regularization for conservatism in the joint action space. If we naively apply conservatism using single-agent offline RL methods to each single-agent MDP (e.g. $D_{KL}( d_i(s, a_i) || d^D_i(s,a_i) )$ term in OptiDICE's objective function or $E_{s \sim D, a \sim \mu}[Q(s,a_i)] - E_{s \sim D, a \sim \pi_D}[Q(s, a_i)]$ term in CQL's policy evaluation loss), each agent ends up optimizing a different objective function, which makes it challenging to ensure convergence to a Nash policy. It is also unclear if the obtained joint factorized policy can effectively prevent the OOD joint actions. In contrast, AlberDICE unifies the objective functions of all agents by adopting *conservatism on joint actions* (i.e. $D_{KL}(d_i(s, a_i) \pi_{-i}(a_{-i}|s) || d^D(s, a_i, a_{-i}) )$), which explicitly prevents the OOD joint actions while providing a convergence guarantee to a Nash policy. Please also see Section A.3 in the Appendix for a detailed analysis on how a naive extension of OptiDICE is problematic in the multi-agent setting. Finally, we present extensive experiments on a wide range of domains and levels of difficulty where avoiding the selection of OOD joint actions and/or convergence to sub-optimal joint policies is challenging. In these environments, we  (1) identify the shortcomings of common approaches such as fully decentralized training and value factorization and (2) introduce the first DICE approach as a principled approach to avoiding distribution shift in Offline MARL.
>
> > **W2. The key point for AlberDICE is Theorem 4.2 which NE can be achieved. However, it is theoretically unclear to me why after the optimization, the inequality between lines 591-592 in the appendix holds. Let's consider an extreme case in the Rock Paper Scissors game. If the dataset only contains draw cases, how does AlberDICE reach NE? Maybe Theorem 4.2 lacks the necessary assumptions.**
>
> The inequality in the proof of Theorem 4.2 holds because, after optimizing, agent $i$ cannot be worse off than before the optimization. Please note that the Rock Paper Scissors game is NOT an MMDP with a common reward structure, but rather it is a *zero-sum game*. In the case of non-cooperative MARL setting like zero-sum game, monotonic improvement cannot be guaranteed in general. In this paper, we are considering the cooperative MARL with common reward setting.
>
> > **Q1. $x$ in eq. 10-14 is sample $(s, a, s’), e_v$ is also a function of $(s,a)$.How eq.14 is state-dependent instead of state-action-dependent?**
>
> In line 175, we mentioned that "the function to learn" is state-dependent. The function that we are optimizing, i.e. $\nu: S \rightarrow \mathbb{R}$, is state-dependent (in contrast to Q-function which is state-action dependent). In (14), we are not optimizing $e$ function but only $\nu_i$ function. We introduced $\hat e_{\nu_i}(x) := r(s,a) - \alpha \log \frac{ \pi_{-i}(a_{-i} |s) }{ \pi_{-i}^D(a_{-i} | s, a_i) } + \gamma \nu_i(s') - \nu_i(s)$ for notational brevity (line 141).
>
> > **(Limitations) The paper does not explicitly mention any limitations of the proposed method.**
>
> Please see Section A.10 of the Appendix.

---

### Official Review · Reviewer_82R1 · 2023-07-07

**Soundness:** 3 good
**Presentation:** 3 good
**Contribution:** 2 fair
**Rating:** 5
**Confidence:** 4

**Summary:**

In this paper, the authors propose an alternative multi-agent offline reinforcement learning algorithm, where each agent is optimized for the best response policy and avoiding the out-of- distribution joint actions. The theoretical part proves the nash policies convergence after the presented training paradigm. The empirical results show promissing results on several well- known environments over existing offline MARL algorithms.

**Strengths:**

● The paper is well-written and easy to follow.

● The theoretical results show the Nash policy convergence which integrates the trackable theory, relaxing the conventional IGM regularization.

● The joint-action problem is novelly addressed in the offline MARL algorithm.

**Weaknesses:**

● The solution of the XOR game shown in the introduction for IGM-based and decentralized methods may not converge to (B,B) from the reviewer's perspective. These two categories of methods will connect the agents' policies by the mixing network for example, which is not
 totally individually be executed.

● The claim from the authors for circumventing the curse of dimensionality in MARL is somewhat overclaimed. The curse of dimensionality also includes the explosion in the joint policy space due to the increasing agent number.

**Questions:**

See the weakness part.

**Limitations:**

In the practical scenarios, the observation and action space may have some noise that is not computable and may obstacle to the presented method.

---

> ### Author Rebuttal · Authors · 2023-08-06
>
> # Response to Reviewer $\text{\textcolor{green}{82R1}}$
>
> Thank you for taking the time to review our paper. Please feel free to let us know if any of our responses below need further clarification. We’d be more than happy to address them during the reviewer-author discussion period.
>
> > **W1. The solution of the XOR game shown in the introduction for IGM-based and decentralized methods may not converge to (B,B) from the reviewer's perspective. These two categories of methods will connect the agents' policies by the mixing network for example, which is not totally individually be executed.**
>
> As we have shown in the experiments, both IGM-based and decentralized methods converge to $(B, B)$ in the XOR Game (see Table 1 in Section 5.1 and the full results in Table 5 of Appendix Section A.7). The mixing network the reviewer refers to is used during training as a mechanism to aggregate the individual Q-functions to represent the global Q-function. However, since we are in the Centralized Training Decentralized Execution (CTDE) setting, the mixing network cannot be used during execution, and the policies must act independently (e.g. using only individual Q for each agent) without any correlation mechanism or centralized information.
>
> > **W2. The claim from the authors for circumventing the curse of dimensionality in MARL is somewhat overclaimed. The curse of dimensionality also includes the explosion in the joint policy space due to the increasing agent number.**
>
> We respectfully disagree with the reviewer that we have overstated our claims of avoiding the curse of dimensionality. AlberDICE offers a principled mechanism to avoid the explosion in the joint action space by using an alternating optimization approach to finding the best response for each agent. This automatically reduces the policy space as well, since, at each optimization step, each agent is solving a reduced MDP (see Line 127-131) that is defined over the *individual* action space. Finally, we have shown in our experiments (especially in the Multi-Robot Warehouse results in Section 5.3) that AlberDICE significantly outperforms other baselines when the number of agents increases and the probability of the agents colliding increases.
> Please let us know if our understanding of this concern is correct and we’d be more than happy to provide further clarifications if necessary.
>
> > **(Limitations) In the practical scenarios, the observation and action space may have some noise that is not computable and may obstacle to the presented method.**
>
> We are slightly confused regarding this statement. It would be very helpful for us if the reviewer can elaborate on this point. For example, is this referring to having noisy observations? If the comment is about the fact that we are considering MMDPs rather than Dec-POMDPs in the paper, please see the footnote for Line 77 in the Background section. To reiterate, we consider MMDPs for simplicity but our algorithm can be extended to partially observable settings, as we show in our experiments.

---

### Official Review · Reviewer_gucm · 2023-07-20

**Soundness:** 3 good
**Presentation:** 3 good
**Contribution:** 3 good
**Rating:** 7
**Confidence:** 3

**Summary:**

The authors consider the offline multi-agent reinforcement learning setting. They provide an example to demonstrate that decentralisation can lead to out of distribution actions at test time. They derive AlberDICE, an offline RL method based on computing alternating best responses to the other agents' policies. Starting from the linear program for an MMDP, they learn a policy by:

1. Pretraining a data policy $\pi^D_{-i}(a_{-i} | s, a_i)$
2. Using the current policy and current data policy to resample the dataset
3. Solve a minimisation problem to find $\nu^*$
4. Extracting the policy from $\nu^*$ using I-projection.

This method prevents out of distribution actions by optimising each agent alternately and assuming the other agents' policies are fixed. A KL penalty is used to ensure that the resulting learned policy stationary distribution stays faithful to the dataset's distribution. By alternately optimising each agent, issues of producing out-of-distribution actions due to simultaneous updates do not arise.

**Strengths:**

Overall I enjoyed reading this paper. In particular:
* The paper evaluates their algorithm on a wide range of domains and performs strongly across most tasks.
* The authors explain the problem setting clearly and point out some problems with naive approaches to the problem.
* The paper is generally well-written and clearly presented.

**Weaknesses:**

While the paper is good overall, there are some weaknesses of the work.
* It would have been nice to see an explicit demonstration that the proposed method reduces out-of-distribution actions in the more complex environments and therefore that the mechanism proposed is the reason for the better performance.
* The evaluations mostly focus on a small number of agents ($\leq 10$).
* Although the authors evaluate on a wide range of domains, they only evaluate on a small number of SMAC maps. This is a well-documented problem with MARL evaluation [1].
* The discussion in lines 125--131 doesn't seem to make clear the presence of the KL penalty term in (2). For example, $\pi^*$ is only the optimal policy *for the MMDP with a penalty as the reward*. Adding a few words in key places clarifying this point would make this section easier to read -- I was a little confused on first reading.



[1] Gorsane, Rihab, et al. "Towards a standardised performance evaluation protocol for cooperative marl." Advances in Neural Information Processing Systems 35 (2022): 5510-5521.

**Questions:**

I have a few clarifying questions.

- How does the method scale to larger numbers of agents? Is the relatively small number of units in each scenario because of computational constraints running this method for more units?

- In equation (4), the $\pi_{-i}(a'_{-i} | s')$ from equation (3) seems to have disappeared. It reappears later in equation (5) so I think this is just a typo. Is this correct or have I missed a detail here?


**Limitations:**

The author discuss the limitations briefly in the appendix. The discussion there seems fine to me but of course more discussion of limitations is always appreciated.

---

> ### Author Rebuttal · Authors · 2023-08-10
>
> # Response to Reviewer $\text{\textcolor{magenta}{gucm}}$
>
> Thank you for taking the time to provide detailed feedback on our work. We are happy to hear that you have enjoyed reading our paper. Please feel free to let us know if any of our responses below need further clarification. We’d be more than happy to address them during the reviewer-author discussion period.
>
>
> > **W1. It would have been nice to see an explicit demonstration that the proposed method reduces out-of-distribution actions in the more complex environments and therefore that the mechanism proposed is the reason for the better performance.**
>
> We conducted additional experiments to quantitatively see that AlberDICE reduces OOD joint actions, compared with baselines in SMAC domains (see the PDF in the general response). In short, we first trained an uncertainty estimator $U(s, a)$ using the dataset $D$. Then, we evaluate $U(s, \pi(s))$ for each $s \in D$ and each algorithm's resulting policy $\pi$ and draw a histogram of the estimated uncertainties. As can be seen in Figure B1 in the PDF, AlberDICE selects OOD joint actions significantly less frequently  than ICQ and OMAR while outperforming the baselines in terms of success rate.
>
> > **W2. The evaluations mostly focus on a small number of agents ($\leq 10$).**
>
> Currently, evaluating on a maximum of 10 agents is considered to be the state-of-the-art in Offline MARL [5-8]. Scaling to $> 10$ agents in an efficient way remains to be an important challenge in MARL in general, as described by Section 1.4 (Challenges of MARL) in [4].  In this work, we focus on the problem of joint action OOD and as we show in our experiments in Section 5 (especially in the Multi-Robot Warehouse domain) the performance gap between AlberDICE and other baselines widens as the *density* of the agents in an environment increases and more coordination is required to avoid sub-optimal behavior such as collisions. We leave the scalability to a large # of agents as future work.
>
> > **W3. Although the authors evaluate on a wide range of domains, they only evaluate on a small number of SMAC maps. This is a well-documented problem with MARL evaluation [1].**
>
> We conducted additional experiments for two more SMAC maps, and please see below the results: 8m_vs_9m (Hard) and 3s5z_vs_3s6z (Super Hard) where AlberDICE still outperforms the baseline algorithms.
>
> | Scenario |      BC     |    ICQ     |  OMAR     |    MADTKD     |  OptiDICE  | AlberDICE |
> | :------------------:| :---------:|:---------:|:----------:|:-----------:|:----------:|:-----------:|
> | 8m_vs_9m (Hard) | 0.48±0.05|  0.12±0.21|   0.45±0.05| 0.14±0.04| 0.47±0.05| **0.67±0.06**|
> | 3s5z_vs_3s6z (Super Hard)|   0.45±0.03|  0.31 ± 0.04 | **0.60±0.05** | 0.18±0.02 | 0.42±0.04| **0.63±0.03**|
>
> However, we would like to point out that the problem raised by the reviewer is written in context to the larger problem that many papers use a single domain (such as SMAC) as their high-dimensional benchmark. On the other hand, we have provided extensive experiments on a wide variety of low to high-dimensional coordination tasks (XOR Game, Bridge, Google Research Football and Multi-Robot Warehouse) which are still challenging for many existing algorithms. Thus, we have focused on 6 (3 Hard, 3 Super Hard) maps which are relatively more difficult than other SMAC maps.
>
> > **W4. The discussion in lines 125-131 doesn't seem to make clear the presence of the KL penalty term in (2). For example,  $\pi^\ast$ is only the optimal policy for the MMDP with a penalty as the reward..**
>
> Thank you for the suggestion. We will make it clear in the final version of the paper to improve readability.
>
> > **Q1. How does the method scale to larger numbers of agents? Is the relatively small number of units in each scenario because of computational constraints running this method for more units?**
>
> The computational cost of our method grows linearly with the number of agents, thus it can solve the problems with a larger number of agents in principle. In our current implementation and computational resources, running experiments with 10 agents took around a few days, thus conducting experiments with tens or hundreds of agents is expected to take a few weeks, which is beyond our computational capabilities. Still, the main focus of this paper is to address the issue of OOD joint actions in offline MARL. To this end, we conducted experiments on an extensive set of MARL benchmarks up to 10 agents, where AlberDICE consistently outperforms the baselines while effectively avoiding OOD joint actions (see the PDF in the general response).
>
> > **Q2. In equation (4), the $\pi_{-i}(a_{-i}’|s’)$  from equation (3) seems to have disappeared. It reappears later in equation (5) so I think this is just a typo. Is this correct or have I missed a detail here?**
>
> $-\sum_{a_i^\prime} d_i(s’, a_i’)$ in (4) is equal to $-\sum_{a_i^\prime, a_{-i}^\prime} \pi_{-i}(a_{-i}^\prime | s’) d_i(s’, a_i’)$, given that $\sum_{a_{-i}} \pi_{-i}(a_{-i} | s') = 1$ since $\pi_{-i}$ should be a valid probability distribution over $a_{-i}$.

---

> > ### Comment · Reviewer_gucm · 2023-08-17
> >
> > Thank you very much for your response to my review, which has answered my questions. I agree with the authors about the evaluation on multiple benchmarks and think the authors have evaluated their method well.

---

> > > ### Author Response · Authors · 2023-08-17
> > > **Thank you for acknowledging our rebuttal.**
> > >
> > > Thank you very much for acknowledging our rebuttal. We are pleased to hear about your positive comments regarding the evaluation of our approach. Please let us know if any further questions or concerns come up and we would be happy to clarify them anytime during the discussion period.

---

### Official Review · Reviewer_s9Ry · 2023-07-26

**Soundness:** 3 good
**Presentation:** 2 fair
**Contribution:** 3 good
**Rating:** 5
**Confidence:** 3

**Summary:**

This paper introduces the AlberDICE framework that targets to address the out-of-distribution challenge (or distributional shift) in off-policy MARL. AlberDICE addresses this challenge by the alternating optimization procedure along with the regularization term. Also, AlberDICE addresses the curse of dimensionality in multi-agent settings with the autoregressive policy architecture. Empirical evaluations show that AlberDICE can find a more optimal solution than baselines in multiple multi-agent benchmark domains.

**Strengths:**

1. Theoretical contribution about showing how AlberDICE converges to a Nash equilibrium.
2. Evaluations are performed across multiple domains, showing the empirical evidence of AlberDICE.

**Weaknesses:**

1. A possible concern about alternatively finding the best response policy while fixing others is that this approach effectively solves a static single-agent problem from each agent's perspective. As a result, in some cooperative settings, this method may learn some sub-optimal joint collaborative policy.
2. Related to #1, the approach of alternatively finding the best response may not work or be effective in competitive/mixed-sum game settings, limiting the scope of the proposed method.

**Questions:**

In addition to my concerns in the weaknesses section, I would like to ask the following questions:

1. I appreciate the intuitive example (Figure 1) in the Introduction. If a centralized critic is employed with a decentralized actor, would the sub-optimality problem still happen?
2. Section 3 states that "when each agent is optimized alternatively, the objective function will always be monotonically improved and convergence is guaranteed." Would there be a proof or reference to this statement?
3. Related to #2, while the monotonic improvement is guaranteed, would convergence to an optimal (Nash) equilibrium be guaranteed?
4. According to Levine et al., 2020, the main challenge of off-policy learning is the distributional shift: a policy is trained under one distribution and it will be evaluated on a different distribution. In the evaluation section, it is unclear whether the learned cooperative policies are evaluated under a different distribution/environment/setting compared to the training dataset.
5. Because the authors focus on cooperative settings, wouldn't it be better to target to converge to a correlated equilibrium instead of a Nash equilibrium as correlated equilibria are a superset of Nash equilibria (i.e., possible to converge to a more optimal solution).

Reference:
Levine et al., Offline Reinforcement Learning: Tutorial, Review, and Perspectives on Open Problems, 2020

**Limitations:**

Yes, the authors have adequately addressed the limitations.

---

> ### Author Rebuttal · Authors · 2023-08-10
>
> # Response to Reviewer $\text{\textcolor{purple}{s9Ry}}$
>
> Thank you for taking the time to provide feedback on our paper. Please feel free to let us know if any of our responses below need further clarification. We’d be more than happy to address them during the reviewer-author discussion period.
>
> > **W1. …in some cooperative settings, this method may learn some sub-optimal joint collaborative policy.**
>
> As discussed in the Limitations section (Appendix A.10), our method that computes a Nash policy does not necessarily converge to the globally optimal joint policy, but instead it can converge to a local optimum. However, we have empirically demonstrated that AlberDICE performs well when just starting from a randomly initialized policy, significantly outperforming baseline algorithms on an extensive set of MARL benchmarks (Penalty XOR, Bridge, RWARE, GRF, and SMAC). Please also note that existing offline MARL algorithms relying either on IGM-principle (e.g. ICQ) or on decentralized training (e.g. OMAR) may struggle to solve even very simple problems (XOR, Bridge), whereas AlberDICE is capable of computing an optimal joint policy for these problems.
>
>
> > **W2. the approach of alternatively finding the best response may not work or be effective in competitive/mixed-sum game settings, limiting the scope of the proposed method.**
>
> In this paper, we primarily focus on the fully cooperative MARL in an offline setting, in line with previous works [5-8]. Our main goal is to tackle the challenge of avoiding OOD joint actions, which can occur even in a fully cooperative setting and also has not yet been precisely addressed in the existing works, to the best of our knowledge. Furthermore, there are currently no standard high-dimensional benchmarks and datasets for general-sum MARL. Thus, we leave the extension of AlberDICE to solving general-sum settings in MARL as future work.
>
> > **Q1. If a centralized critic is employed with a decentralized actor, would the sub-optimality problem still happen?**
>
> If the centralized critic is employed without limiting expressiveness, it may not suffer from the sub-optimality problem in principle. Still, it suffers from the exponential complexity with respect to the number of agents (i.e. curse of dimensionality), which makes it not scalable. Thus, to improve scalability, existing offline MARL algorithms such as ICQ [8] adopts (IGM-based) value decomposition structures with limited expressiveness for the centralized critic. Consequently, as can be seen in Table 1, ICQ, a representative offline MARL algorithm using a centralized critic and decentralized actor, suffers from the sub-optimality problem. Please see the Introduction (Lines 26-52) for an illustrative explanation for how ICQ can converge to sub-optimal OOD joint actions.
>
> > **Q2. Section 3 states that "when each agent is optimized alternatively, the objective function will always be monotonically improved and convergence is guaranteed." Would there be a proof or reference to this statement?**
>
> Yes, this is shown in the proof of Theorem 4.2 in Appendix A.4. A reference to Section 4 is indeed appropriate here and we will include this in the final version of the paper.
>
> > **Q3. Related to #2, while the monotonic improvement is guaranteed, would convergence to an optimal (Nash) equilibrium be guaranteed?**
>
> Convergence to a Nash policy is guaranteed, as shown in Theorem 4.2. However, please note that this Nash policy may not be optimal, as we discuss in the Limitations in Section A.10 in the Appendix. The outcome of the iterative best response depends on the starting point (region of attraction of each Nash policy) and is, thus, generally not guaranteed to find the optimal Nash policy [1]. This is the notorious equilibrium selection problem which is an open problem in games with multiple equilibria, even if they have common reward structure (See Open Questions in [2]). Nevertheless, good equilibria tend to have larger regions of attraction and practical performance is typically very good as demonstrated by our extensive experiments.
>
> > **Q4. In the evaluation section, it is unclear whether the learned cooperative policies are evaluated under a different distribution/environment/setting compared to the training dataset.**
>
> We followed the evaluation protocol of standard offline (MA)RL [3]. The training dataset was collected by (multiple) data-collection policies in MMDP (or Dec-POMDP). Then, each offline MARL algorithm optimizes its target policy only using the offline training dataset, where distribution shift occurs since the target policy will be deviated from the data distribution to maximize the rewards. Finally, the learned (cooperative) policies are evaluated under the same MMDP (or Dec-POMDP) environment as in the data-collection procedure. We are not considering multi-task RL or domain transfer in this paper. Please refer to Section A.6 in the Appendix for a detailed explanation of how the dataset was constructed for all environments.
>
> > **Q5. [Why not Correlated Equilibria]**
>
> The problem with correlated equilibria is that they are defined in the joint policy space, i.e., they can be non-factorizable (unless they are Nash equilibria) and hence, problematic for decentralized execution. For instance, the distribution 0.5/0.5 over (A,B) and (B,A) in the XOR game (Figure 1) is an (optimal) correlated equilibrium, but such a policy is not in a form that allows decentralized execution. Naively making a factorized policy from the correlated equilibria policy by considering its marginal distributions (0.5,0.5 for each player) leads to a (very) suboptimal set of policies for decentralized execution. In fact, maintaining a factorizable policy throughout its iterations, and ultimately, finding a Nash equilibrium (suitable for decentralized execution), rather than a correlated or coarse correlated equilibrium is one of the major contributions of AlberDICE.

---

> > ### Comment · Reviewer_s9Ry · 2023-08-14
> >
> > I appreciate the authors' clarifications, and these answer my questions. After reading other reviewers' comments//authors' responses and the additional experiments performed during the rebuttal, I would like to maintain my score but with more toward acceptance because AlberDICE shows more strong empirical results than its main baselines (e.g., OptiDICE) across many domains (e.g., Table B1).

---

> > > ### Author Response · Authors · 2023-08-16
> > > **Thank you for the response.**
> > >
> > > Thank you very much for your prompt response to our rebuttal. We appreciate your positive remarks especially regarding our strong empirical results. Please let us know if there are any further questions or concerns and we would be happy to clarify them anytime during the discussion period.

---

### Author Rebuttal · Authors · 2023-08-10

# General Response

We thank all reviewers for taking the time to provide constructive feedback on our paper. We are encouraged that they find the following strengths in our work:

- Clear presentation of the problem with OOD joint actions and the failures of previous approaches (Reviewers $\text{\textcolor{magenta}{gucm}}$, $\text{\textcolor{green}{82R1}}$)
- Novel approach to solving the joint action OOD problem (Reviewers $\text{\textcolor{green}{82R1}}$, $\text{\textcolor{blue}{5Yz6}}$)
- Theoretical contribution showing convergence to Nash policies (Reviewers  $\text{\textcolor{green}{82R1}}$, $\text{\textcolor{purple}{s9Ry}}$, $\text{\textcolor{blue}{5Yz6}}$)
- Strong empirical performance over baselines on a wide range of MARL environments (Reviewers  $\text{\textcolor{magenta}{gucm}}$, $\text{\textcolor{purple}{s9Ry}}$ $\text{\textcolor{green}{82R1}}$, $\text{\textcolor{red}{aFwQ}}$, $\text{\textcolor{blue}{5Yz6}}$)
- Well-written and easy to follow (Reviewers $\text{\textcolor{magenta}{gucm}}$, $\text{\textcolor{green}{82R1}}$, $\text{\textcolor{red}{aFwQ}}$)

Since the reviewers raised concerns which were mostly different, we reply to each reviewer separately. We also provide additional experimental results in the attached single-page PDF which include the following:

- Demonstration of AlberDICE reducing OOD joint actions in complex domains (Reviewer $\text{\textcolor{purple}{s9Ry}}$)
- SMAC results on 2 additional maps (Reviewer $\text{\textcolor{purple}{s9Ry}}$)
- Ablation results (Reviewer $\text{\textcolor{blue}{5Yz6}}$)

Please let us know if further clarifications are needed as we would be more than happy to answer them during the reviewer-author discussion phase.

Finally, we list below the references which are used throughout the responses.

## References

[1] Multiagent Value Iteration Algorithms in Dynamic Programming and Reinforcement Learning (Bertsekas 2020)

[2] Global Convergence of Multi-Agent Policy Gradient in Markov Potential Games (Leonardos et, al. ICLR 2022)

[3] D4RL: Datasets for Deep Data-Driven Reinforcement Learning (Fu et, al. 2021)

[4] Multi-Agent Reinforcement Learning: Foundations and Modern Approaches (Albrecht et, al. Preprint MIT 2023)

[5] Offline pre-trained multi-agent decision transformer (Meng et, al. Arxiv 2022)

[6] Plan better amid conservatism: Offline multi-agent reinforcement learning with actor rectification (Pan et, al. ICML 2021)

[7] Offline multi-agent reinforcement learning with knowledge distillation (Tseng et, al. NeurIPS 2022)

[8] Believe what you see: Implicit constraint approach for offline multi-agent reinforcement learning (Yang et, al. NeurIPS 2021)

---

### Comment · Area_Chair_L2q1 · 2023-08-18
**Urgent request to engage**

Dear Reviewers 82R1and 5Yz6,

Please urgently engage in the discussion phase, since it is coming to the end.
The authors provided detailed responses to the points raised as well as additional experimental evaluations.
It is absolutely vital that these datapoints are carefully considered in the process.

Many thanks!

AC

---

### Decision · Program_Chairs · 2023-09-21

**Decision:**

Accept (poster)

**Comment:**

The authors present a novel method, AlbertDICE, for training offline policies in fully cooperative multi-agent RL settings.

Specifically, the authors apply a coordinate ascent type strategy, but optimising one agent at a time.
They also provide a novel mechanism to deal with the ood problem of action selection in offline RL and provide theoretical guarantees.

Overall, the reviewers agreed that the paper makes valuable contributions and that the empirical results are impressive, in particular on the super-hard SMAC maps.

Based on taking a look at the paper I have a few suggestions for improvement:
1) The paper introduces MMDPs (fully observable) but evaluates on SMAC (Dec-POMDP). It would be great to resolve this inconsistency.
2) SMAC has largely become obsolete, at least for online RL. Time permitting, it would be great to test on SMACv2 instead (https://sites.google.com/view/smacv2).
3) Authors should also take the suggestions for the reviewers to hard and include the new results and ablations from the rebuttals.

Assuming those changes I recommend that the paper is accepted for publication if there is space.